# A personalized reinforcement learning recommendation algorithm using bi-clustering techniques

**Muhammad Waqar, Mubbashir Ayub** [ID] *

Department of Software Engineering, University of Engineering and Technology, Taxila, Pakistan

* mubbashir.ayub@uettaxila.edu.pk

## Abstract

Recommender systems have become a core component of various online platforms, helping users get relevant information from the abundant digital data. Traditional RSs often generate static recommendations, which may not adapt well to changing user preferences. To address this problem, we propose a novel reinforcement learning (RL) recommendation algorithm that can give personalized recommendations by adapting to changing user preferences. However, a significant drawback of RL-based recommendation systems is that they are computationally expensive. Moreover, these systems often fail to extract local patterns residing within dataset which may result in generation of low quality recommendations. The proposed work utilizes biclustering technique to create an efficient environment for RL agents, thus, reducing computation cost and enabling the generation of dynamic recommendations. Additionally, biclustering is used to find locally associated patterns in the dataset, which further improves the efficiency of the RL agent's learning process. The proposed work experiments eight state-of-the-art biclustering algorithms to identify the appropriate biclustering algorithm for the given recommendation task. This innovative integration of biclustering and reinforcement learning addresses key gaps in existing literature. Moreover, we introduced a novel strategy to predict item ratings within the RL framework. The validity of the proposed algorithm is evaluated on three datasets of movies domain, namely, ML100K, ML-latest-small and FilmTrust. These diverse datasets were chosen to ensure reliable examination across various scenarios. As per the dynamic nature of RL, some specific evaluation metrics like personalization, diversity, intra-list similarity and novelty are used to measure the diversity of recommendations. This investigation is motivated by the need for recommender systems that can dynamically adjust to changes in customer preferences. Results show that our proposed algorithm showed promising results when compared with existing state-of-the-art recommendation techniques.

## I. Introduction

Recommender systems enhance user experience by providing personalized relevant content from the pool of vast majority of digital data [1]. This helps in increased user satisfaction and

**Data Availability Statement:** All relevant data for this study are within the paper.

**Funding:** The author(s) received no specific funding for this work.

**Competing interests:** The authors have declared that no competing interests exist.

engagement toward the utility of digital platforms, which ultimately increases productivity and customer engagement for a particular platform [2]. Collaborative filtering and content-based filtering are widely used major techniques of recommender systems. However, major limitations of these techniques are adaptability, lack of personalization and diversity [3]. These techniques are static in nature, which means that they cannot adapt to changing user's preferences. Moreover, with increased size of data, performance of these techniques decreases gradually [4]. In recent years, advancement in machine learning techniques particularly in RL field has led to possibility of getting dynamic recommendations [5], which tends to change with changing users' preferences. However, RL based recommender systems are computationally very expensive, when it comes to dealing with large datasets [6]. Clustering algorithms are oftenly used to extract global patterns/associations within the dataset. This can led to a situation where many of the local interactions of users across the dataset remain unexplored [7].

In this work we are combining the benefits of biclustering with power of RL to generate personalized recommendations. Biclustering enables us to generate clusters of users-items based on pattern based interaction, thus, extracting localized information from the dataset. RL enables us to learn and adopt to changing user's preference over time to optimize policy for generating recommendations. The fundamental aspect of the proposed research is to combine the complementary nature of biclustering and RL in context of recommender systems. Thus, generating recommendations that are more personalized, diverse and accurate. As, RL based algorithms generally consume extensive resources during execution. We placed Biclusters on a squared grid as a cost effective solution for traversing through environment.

Many biclustering algorithms are proposed in literature and are being used by bioinformaticians for exploration of genes and conditions hidden correlations. So, our first objective in this work to determine that can we use these biclustering algorithms in the recommender system domain to determine hidden correlations of users with items? Also can any biclustering algorithm be used for this purpose or cannot be used? In order to find answer of these questions we used eight different biclustering algorithms and found that not all algorithms can be used to solve recommendation problem. More detail of these algorithms will come in next sections of this paper.

To model recommendation problem as an MDP problem, we first applied eight biclustering algorithms to user-item ratings matrix. After applying eight Biclustering algorithms, it is observed that performance of all algorithms is not similar, saying it more specifically, they generate different type of biclusters from the same data. The biclusters generated by these algorithms are measured in term of quality on the basis of values contained in them. Top $n^2$ most suitable biclusters are selected, which are then placed on $n \times n$ squared grid. Each bicluster represents a state on n*n squared grid containing a set of rows representing users while set of columns representing items. Proposed RL agent moves over the $n \times n$ grid to get personalized recommendations. Action space is designed to include four actions i.e. left, right up and down. Start state in the recommendation process is computed by an intra triangular similarity. Deterministic transition function follows $\in$-greedy policy to balance exploration and exploitation within grid. Reward function is measured as overlapping of users in current and prospective next state. If agent is not receiving recommendation of any new items or getting off the grid, then this state is declared as goal state of the agent. Objective of RL agent is to maximize cumulative reward termed as return, which denotes how much user is satisfied with recommended items. Moreover, contrary to existing RL based recommender systems, we developed a novel approach to predict ratings for recommended items. This enables us to calculate MAE and RMSE for the recommended items' rating. This numerical calculation of predicted ratings for RL based recommender systems was unprecedented as per our knowledge so far. In reinforcement learning (RL) based recommender systems, rating prediction can improve quality of

recommendations' offering customers specific suggestions based on their preferences. This makes it possible to evaluate recommendations in greater detail, and, in turn, improving customer's satisfaction. Moreover, to evaluate efficiency of proposed algorithm across all dimensions, some new evaluation metrics like diversity, intra list similarity, personalization and novelty are also used.

Rest of the paper is organized in the following sections, section II is related work section that put insights on the work done by other researchers in the field of RL based RSs. Section III discusses material and methods used for the proposed research work. Section IV describes experimental procedure. Section V discusses observed experimentation results on the given datasets (ML-100K, ML-latest small and FilmTrust). Section VI concludes our work and gives some future directions.

## II. Related work

Recommender systems are software tool that are used to facilitate users in finding items of interest. On one hand, these systems can be used to predict user preferences for a particular unknown item and on the other side, they are used to cater long tail problem. The long tail problem [8] occurs when there are many useful items which remain unvisited from the users due to availability of large number of items. Recommender system is helpful to find relevant items in huge pool of information for a particular user. In a broader term, recommender system can be divided into following categories on the basis of method of recommendations.

### Content based filtering

Content based filtering recommend items to a user that have similar content/features to the items that are previously rated by the active user. The basic rule of content-based filtering [9] is to create feature vector for particular items and store them as a description in user profile. Whenever an item is to be recommended to a particular user, the description of new item is matched with the description given in user profile and those items who have high similarity with the users are then recommended. In content-based filtering [10] many of the similarity measures like Pearson correlation, cosine etc. are used to find matching items and users. In recent times, statistical modeling is also being done that can be used to construct model that can learn from user's past history.

### Collaborative filtering

Collaborative filtering systems [11] uses wisdom of crowd to recommend items which are liked by most similar users of user seeking recommendations (active user). Collaborative filtering [12] can be divided into two categories, item-based collaborative filtering and user-based collaborative filtering. In item-based collaborative filtering, similar items are found for the active user on the basis of items which he has liked in the past. In user-based collaborative filtering [13], user will receive recommendations for the items liked by the similar users. Cosine similarity, adjusted cosine similarity, Pearson correlation and Jaccard index are some of the most famous similarity measures that are used to compute similarity between users and items.

### Clustering based systems

One major problem that may arise while dealing with recommender system is that recommendation process involves huge amount of dataset which may bring about challenges like data sparsity and scalability. Clustering techniques [14] helps to reduce scalability and sparsity problem that arises due to huge volume of dataset for recommender systems. The main idea

for clustering underlines that dataset consists of various entities. Each entity is characterized by set of features; the general idea is to combine items having similar features into one group. This clustering can be done in various way. Partitional, hierarchical co clustering, adaptive and fuzzy clustering are some of the most famous techniques [15] for applying clustering to given dataset. However, finding an appropriate clustering algorithm for a given dataset is a difficult job due to varying nature of dataset and its statistical distribution [16]. Content based filtering uses output of clustering algorithms to generate recommendation using features of cluster that appear to be most similar to the active user for which recommendation is to be made. Apart from that, various other clustering techniques [17] are being used for hybrid recommendation systems by merging various users and items into various clusters based on certain features.

## Computational intelligence based systems

These systems aims to construct recommendation models to generate prediction for unseen items [18]. The common techniques which are used to construct recommendation models includes matrix factorization, artificial neural network, clustering techniques, genetic algorithms and fuzzy set techniques. Bayesian classifier [19] is used as probabilistic model for recommendation of items to active user. Bayesian classifier is used as model-based approach to generate predictions. In Bayesian network, each node represents an item and state represents vote value. Furthermore, hierarchical Bayesian network have also been used to combine benefits of content and collaborative filtering.

Artificial neural networks have been deployed in recent past in the field of recommendation system. The major advantages of neural network are that it helps to achieve higher accuracy. The other aspect regarding neural network is that it is flexible and can be used to include multiple factors for recommendation purposes. The general idea of using neural network [20] for recommendation process is to create embedding vectors for items and users. The dot product between item vector and user vector will give us the similarity measures and likelihood of how much a user will like a particular item. Furthermore, in order to enrich recommendation process more of innovative features like gender, age and city can be used for recommendation.

Genetic algorithms [21] are also used in intelligent recommender systems that can recommend relevant items to various users. The general idea is to construct population of possible recommended items. Afterward, some objective function is applied to check the fitness of possible items and then items having high fitness values are used to create new population using mutation, cross over or various other innovative techniques. Users are constantly provided with items for recommendation which have high fitness values. Fuzzy logic techniques [22] have been extensively used in recommender system in order to capture uncertainty, vagueness and ambiguity in recommendation process. Fuzzy logic is used to predict relevant items to users based on fuzzy inference system. Fuzzy logic underlines that a user can be a part of multiple groups based on value of membership function. Higher membership function values highlight the more contribution of particular item or user in fuzzy set. The similarity among users or items can be measured is also measured using membership function's value of item or user vectors.

## Biclustering based recommender systems

Biclustering technique have been extensively [23] used in the domain of bioinformatics where they are used to identify subsets of genes and their conditions. However, their usage in the field of recommender systems have gained popularity in recent times due to their ability to generate diverse and accurate recommendations [24]. Traditional recommender systems often fail to capture user preference and items features simultaneously. This results in generating

recommendations that lack diversity and personalization. Biclustering based recommender system helps to capture cohesive patterns of users and items by simultaneously placing them in separate sub groups [25]. This helps in generation of quality recommendations. Many studies [26] have demonstrated the effectiveness of biclustering in enhancing recommendation accuracy, especially in scenarios where users and items exhibit varying preferences across different contexts or time periods. However, selection of appropriate biclustering algorithms in context of recommender system, their application in regard of recommender system and high computational cost associated with them still remain open challenges for researchers to explore.

## RL based recommender systems

RL based recommender systems [27] considers recommendation as a sequential decision-making process. In this type of recommender systems, there is an agent that will iterate through the environment with effort to maximize the commutative reward. The main idea would be to explore environment [28] in context of some reward function so that relevant items depicting high satisfaction level for user can be recommended. However, the problem that becomes a major hurdle for RL based algorithms are huge state space and possible action 's transitions [29]. There is a lot of research going on to reduce the complexity of RL based recommendation algorithms. RL is being used in recommendation systems since 1999, however the research was very limited. The main reasons for it were, non-availability of standard datasets and in-efficient computational machines, absence of standard evaluation metrics and testing environment to evaluate RL based recommendation algorithms. After 2018, there is a burst in research regarding the application of RL and deep RL in recommender systems.

Each algorithm has its own advantages and disadvantages. Dynamic programming is one of the algorithms that has been used in RL. This algorithm builds [30] Q learning table to learn about the environment. But, it is computationally expensive and becomes infeasible for large state space. In contrary to dynamic programming methods, Monte Car [31] method does not require much knowledge about the environment and only need section of experience to learn about the environment. However, one major drawback of this method is that it converges slowly as they update value function after completing the whole episode. Temporal difference is one famous algorithm of RL based recommendation system which is model free and need minimum computation. Q learning and SARSA [32] are two of the famous temporal difference algorithms. The major drawbacks of these methods are they need extensive computation to complete execution. In recent times, deep reinforcement-based learning [33] has gained a lot of popularity due to its efficiency. This algorithm uses experience relay to store information about the environment. Moreover, this experience is used to update weights in training. This weight updating occurs in batches and for the calculation of error derivative, simplified reward function is used. Actor critic method is one of the famous methods of deep RL which criticize agent for the policy generated for the given state action pair and give feedback on the policy chosen by the agent. Monte Carlo is another famous deep RL based algorithm which is stochastic in nature and uses policy gradient to update weights. However, it is slow and takes a lot of time to learn from the environment. The reason why RL based recommendation system becoming popular is because of its dynamic nature and flexibility to adopt to user's interest. RL based algorithms can be used to create an environment that can effectively represent user's latent state and adapt to it. RL can be used to optimize reward in such a way that it can best represent user satisfaction and can be used to learn about long term interest of a user. Recent work [34] on RL based recommender system have effectively modeled recommender system as a MDP problem. In this work, user-item voting matrix have been used as an environment

for RL agent where agent traversed on the basis of custom reward function. However, major limitations of the aforementioned work were that it unable to predict ratings for the recommended items. Moreover, reward function is not suitable for the required problem. Furthermore, quantitative and qualitative analysis of biclustering algorithms and their relative impact on performance of RL agent was missing. These findings underline that there are many limitations and open research areas which are yet to be explored in the field of RL based recommendation algorithms.

The major drawbacks in RL based recommender system [35] are that almost all recommendation system recommend single item which may become very expensive as RL agent has to explore environment before recommending a certain item. Moreover, large action space within large state space, partial observability of user's activities, noisy reward and complexity in designing an effective reward function that can best represent user's preference are the bottleneck for RL based recommender systems. Furthermore, designing an optimal policy that can give best recommendations in an effective way is still a challenging task. Most notable limitation is that there is no proposed mechanism present in the domain of RL based recommender system to predict ratings for the recommended items.

To solve large state space bottleneck, we used Biclustering technique to cluster subset of similar items and users in user-item voting's matrix. The generated biclusters are also helpful in identifying local patterns within the dataset. These biclusters are then mapped to an $n{\times}n$ squared grid. This grid acts an environment for RL agent where each cell/state in the grid corresponds to a bi-cluster. Biclustering helped us in reducing the large state space problem of RL algorithms. The order of bicluster placement on grid is very important. We know that size of grid should be limited to avoid high computation, for that reason $n^2$ number of biclusters should be selected that are to be placed on $n{\times}n$ grid. The quality of these biclusters is measured using proposed quality score. Top $n^2$ biclusters are selected on the basis of the quality measure and then placed on grid in a cantor diagonal manner. RL agent traverse through the grid and learn about optimal policy using customized reward function resulting in generation of recommended items. The proposed method is evaluated using three publicly available datasets. Detailed working of our proposed method is presented in methodology section

## III. Material and methods

Our proposed method consists of several steps, out of which data preprocessing is first step. All steps are explained in coming subsections.

### A. Data preprocessing

We transformed input data into user-item ratings matrix as shown in Table 1. The main reason to create such structured representation of input data is to make it suitable for the input requirements of Biclustering algorithms. During the construction of user-item votings matrix, those items which are not voted by any user are given "—" (Null) values. Biclustering

**Table 1. Example user-item voting's matrix.**

|  | Item1 | Item2 | Item3 | Item4 |
|---|---|---|---|---|
| User1 | 3 | - - - | 4 | 2 |
| User2 | 3 | - - - | - - - | 1 |
| User3 | 5 | 5 | 4 | 2 |
| User4 | 5 | - - - | 5 | 3 |
| User5 | 2 | 1 | - - - | 5 |

**Table 2. Binary form of Table 1.**

|          | Item1 | Item2 | Item3 | Item4 |
|----------|-------|-------|-------|-------|
| **User1** | 1 | 0 | 1 | 1 |
| **User2** | 1 | 0 | 0 | 1 |
| **User3** | 1 | 1 | 1 | 1 |
| **User4** | 1 | 0 | 1 | 1 |
| **User5** | 1 | 1 | 0 | 1 |

algorithms work only on binary data. To do binary transformation, user-item votings matrix is converted into binary matrix using threshold of 1 which means that for those items who have voting greater than equal to 1, is given a binary value of 1. For items having null value are given binary value 0. This binary transformation is mathematically expressed in Eq (1).

$$b = \begin{cases} 1 \ for \ V_{i,j} \geq 1 \\ 0 \ for \ V_{i,j} = null \end{cases} \tag{1}$$

Where $b$ denotes corresponding binary representation of $v_{i,j}$ and $v_{i,j}$ denotes user $i$ voted value on a certain item $j$. Table 2 shows binary representation of input matrix using Eq (1).

## B. Bi-Clustering of data

Clustering is a technique which is used to group similar data into specific clusters using some distance or similarity measures. The basic aim of clustering is to identify hidden patterns within dataset based on certain similarity. Clustering algorithms work on single data dimension [36] either rows or columns but not both. Moreover, for these clustering algorithms, a single user can only exist in one cluster. The overall effect of these limitations is that clustering algorithms can only detect global patterns residing within dataset. They are not good at identifying local patterns (subset of rows and columns) in a dataset. Biclustering algorithms [37] on the other hand, possesses capacity to simultaneously group subset of rows and columns, thus extraction of local patterns residing in a dataset is possible.

Biclustering algorithms Bibit [38], BiMax [39], iterative signature clustering [40], Chengchurch [41], Plaid [42], Large average submatrix [43], Bipartite [44], Qubic [45] and many more, have been widely used in the field of bioinformatics to analyze gene expression data, identifying subsets of genes and conditions that meet a specific criteria. However, their usage in the field of recommender system is very limited. The python biclustering package of Padilha [46], Biclust package of R language [47] and biclustering tool Bideal [48] have been used for the generation of bilcusters from binary matrix. for the implementation of biclustering algorithms. Our proposed RL environment is composed of fixed size (6×6) grid, thus, minimum of 36 biclusters are required to fill the grid. For biclustering algorithms whose generated biclusters are greater than 36, we have sorted top 36 most similar biclusters using Jaccard similarity which will be explained later. However, if the generated biclusters are less than 36, we have decomposed generated biclusters into sub-clusters until we get bicluster number up to 36. For decomposition, we have given larger generated bicluster as input to current biclustering algorithm, which then broken down large bicluster into smaller biclusters. The aforementioned biclustering algorithms are applied to given datasets using Bideal, Biclust and padilhabiclusters generating tools. The aim was to explore hidden local relationship between user and items. These experiments involve the comprehensive analysis of biclustering results, assessing the quality of discovered patterns, and evaluating the effectiveness of algorithms in capturing

latent structures within the most used recommendation datasets. The experimentation was done to analyze biclusters, their quality and corresponding effectiveness to identify local patterns within dataset. Each algorithm is applied to each of the three datasets used in proposed work and their corresponding performance is being evaluated. After the experimentation it was observed that apart from Bibit, BiMax, iterative signature clustering, bipartite and large average submatrix, other biclustering algorithms that includes Plaid, Qubic and Cheng-church showed unusual behavior. It was observed that Plaid and Qubic generated a single very large bicluster which is not suitable for our requirements. Cheng-church generated biclusters that are very large in number, with size of each bicluster almost same as of the input dataset. For these three algorithms (Plaid, Qubic, Cheng-church), we observed that generated biclusters are very few in number along with huge size, so benefit of biclustering is undermined here as local patterns are not being extracted with such big biclusters. The main reasons of underperformance for plaid and Cheng-church algorithms may be highlighted by the fact that these algorithms do not work well with sparse data as most recommendation datasets are more than 90% sparse. Furthermore, these algorithms are very sensitive to noise and may result in generation of large biclusters to account for variability [49]. The objective functions of binary factorization model for plaid and probabilistic generative model for Cheng-church generally favor large biclusters. For Qubic algorithm the nature of algorithm is such that it is greatly influenced by the size of the dataset. As our used datasets are relatively large in size as compared to datasets used in Bioinformatics domain, the generated biclusters of Qubic are very big in size. Furthermore, the default granularity parameter for the Qubic aims to identify large biclusters and objective function of Qubic also prefers large biclusters.

In the light of above discussion, we have sebsequently selected five biclustering algorithms namely Bibit, BiMax, Large average submatrix, Iterative clustering and bipartite biclustering algorithms for recommendation task. These algorithms suit best according to our requirements of extracting efficient and enough local patterns from the given dataset. In the following discussion, we will explain working mechanism of these biclustering algorithms.

**I. Bibit biclustering algorithm.** Bibit biclustering algorithm is used in domain of bioinformatics, where it is applied to identify gene expression data. It works on binary data, extracts coherence/consistent patterns by identifying 1's across columns in subsequent rows. Each row represents gene while each column represents condition. It emloys bottom up approach by starting with a single row and then merging rows and columns iteratively to costruct bigger bicluster. Bibit's principles are being adpated to handle recommendation datasets mentioned in this work. The given dataset is being converted into user-item rating matrix where each row represents a single user while column represents an item. The value within these matrix is actually rating given to the corresponding item by respective user. To transform this matrix into binary matrix, binary transformation is applied.

**II. BiMax biclustering algorithm.** Just like Bibit, bimax algorithm also specializes to work on binary data. It recursively adds coherent biclusters by identifying similar patterns of 1's and 0's residing within rows and columns across the given dataset. BiMax uses truncation and rotation to refine generated biclusters. Moreover, quality metrics are used to evaluate coherence of generated biclusters. BiMax deploys holistic approach to generate biclusters where it considers whole dataset as a single matrix. Afterward, through pruning and merging new biclusters are generated.

**III. Bipartite biclustering algorithm.** Bipartite biclustering algorithm extracts coherent patterns between two variant sets. One set consists of rows and the other set consist of columns. It recursively adds new rows and columns to both sets. Just like BiMax, it refines biclusters by pruning and rotation. The initial two sets for bicluster generation are selected on random basis. Quality measures are deployed to judge the coherence level of biclusters.

Bipartite biclustering algorithm has been applied in diverse domains such as in bioinformatics to study correlation between gene and subsequent biological functions. Moreover, it can be used to extract hidden interactions between customers and products. One important point regarding bipartite biclustering algorithm is that unlike BiMax and Bibit, it can work on non-binary data. For this reason, original user-voting matrix is given as an input.

**IV. Large average value biclustering algorithm.** In any data matrix, bicluster represents subset of rows and columns which are extracted from the original data. The average of all the values present in the submatrix can be calculated by taking mean of all the values. In context of large average value biclustering algorithm, coherent bicluster means that all those elements of original matrix which share large average values. These high average values can be used to discover coherent patterns and functional relationships with data. For the given problem of recommender system, the given biclustering algorithm detects large/consistent average submatrix from user-item rating matrix to extract possible hidden relations from the data.

**V. Iterative signature biclustering algorithm.** The Iterative Signature Algorithm (ISA) is a biclustering algorithm designed for gene expression data analysis. It starts with seed genes and conditions, iteratively expands biclusters by adding similar genes and conditions, extracts characteristic expression patterns (signatures), and evaluates bicluster quality. ISA aims to discover transcriptional modules—coherent groups of genes with similar expression under specific conditions. It iteratively refines biclusters, accommodating noisy data and capturing overlapping patterns. ISA output offer insights into gene regulation and biological pathways, making it a valuable tool in understanding co-regulation and functional gene groups in complex biological systems.

## C. Ordering and sorting of biclusters

Now we have to order and sort generated biclusters. One way to order and sort biclusters is by measuring quality of biclusters. Quality of a bicluster measure homogeneity of values within the bicluster. Researchers have proposed several quality measurement functions in literature to measure the quality of biclusters. These includes Variance (VAR) [50], Scaling Mean Square Residue (SMSR) [51], Relevance Index (RI) [52] and correlation based measures such as PCC. However, selection of an appropriate quality measure depends on properties of data and type of analysis needed to be performed in order to evaluate biclusters' coherence, significance and relevance. In proposed work, for the creation of RL environment we want to sort biclusters in such a way that similar biclusters are ordered in close proximity. So here we are more interested in measuring similarity of biclusters in contrast to quality of biclusters. As columns in the bicluster represent items, so biclusters having maximum number of columns are best suited for our problem as they can give maximum items that can be recommended to a user. One major problem that should be addressed here is that there may be a possibility that a bicluster may have maximum number of columns but users have not rated enough items in it. This will not give us enough recommended items as most values in bicluster are 0's as user has not rated these items. In this regard, those biclusters should be chosen where users have rated enough items, so that efficient recommendations can be generated. In a nutshell, measuring the similarity of bicluster should be two fold: (i) Identification of a pivot bicluster having maximum columns with high frequency of rated items (ii) Computing Jaccard similarity of all other biclusters with pivot bicluster and sort top-$n^2$ biclusters in descending order of similarity score.

Procedure for selection of pivot bicluster is as follows:

Let $B$ donates a bicluster with dimensions $m \times n$, where $B(i, j)$ is a non-zero value for $1 \leq i \leq m$ and $1 \leq j \leq n$. We want to find a pivotal bicluster $S$ that maximizes both the number of columns

and the number of non-zero values. This can be represented as in Eq (2).

$$S = \{B(i,j)|1 \leq i \leq m, j \in C(S), and\ B(i,j) > 0\} \tag{2}$$

Where

- $S$ represents a bicluster with the maximum number of columns with high frequency of non-zero values.

- $i$ iterates over the rows from 1 to $m$.

- $j$ belongs to the set of column indices $C(S)$, where $C(S)$ represents the set of columns that

- maximize the number of non-zero values in the submatrix.

- $B(i,j) > 0$ ensures that only cells containing non-zero values are included in the bicluster

In order to find $S$, we calculate number of columns of each bicluster and count non zero entries within that bicluster. After adding these two values, score for each bicluster is calculated as shown in Eq (3)

$$Score = m + \sum_{i=1}^{n} \sum_{j=1}^{m} Count(B(i,j) > 0) \tag{3}$$

- $n$ represents total number of rows in a bicluster

- $m$ represents total number of columns in a bicluster

- $B(i,j)$ represents non zero values of a bicluster having row $i$ and column $j$ where $1 \leq i \leq n$ and $1 \leq j \leq m$

After calculating a score for each bicluster using Eq 3, we selected a bicluster as a pivot bicluster $S$, which possess maximum score as given in Eq (3). Afterward, all other generated biclusters are matched with the pivot bicluster user and item set using Eqs (4) and (5) and then sorted in descending order according to their similarity score. Thus top $n^2$ biclusters that can fit to a fixed size squared grid are selected from that similarity list.

$$Sim_{(U_i,U_p)} = \frac{|U_p \cap U_i|}{|U_p \cup U_i|} \tag{4}$$

$$Sim_{(I_i,I_p)} = \frac{|I_p \cap I_i|}{|I_p \cup I_i|} \tag{5}$$

In Eqs (4) and (5) $U_i$ denotes user set of $i^{th}$ bicluster, $U_p$ denotes user set of pivot bicluster, $I_i$ denotes item set of $i^{th}$ bicluster and $I_p$ denotes item set of pivot bicluster. Hence total similarity measure for $n^{th}$ bicluster can be given by Eq (6) as:

$$Sim_{(Total)} = Sim_{(U_i,U_p)} + Sim_{(I_i,I_p)} \tag{6}$$

- $Sim_{(Total)}$ = Total similarity with pivot bicluster

- $Sim_{(user)}$ = User set similarity of bicluster w.r.t pivot bicluster

- $Sim_{(item)}$ = Item set similarity of bicluster w.r.t pivot bicluster

## D. Placement of biclusters on a squared grid

In this section, we will detail out on how to place sorted $n^2$ biclusters on a $n \times n$ square grid. After generating biclusters from a given dataset and with respective similarity evaluation, the next task was to place biclusters into $n \times n$ square grid, which will act as an environment, so that RL algorithm can work on it. However, we faced two challenges here.

**Challenge#1:** How to select the number of biclusters for the given grid? These biclusters are measured in term of their similarity and top $n^2$ biclusters are selected. The similarity measure defined in Eq 6 underlines that every time top $n^2$ biclusters will be selected. Moreover, we cannot increase the size of a grid too much as it will make the computation very expensive with respect to RL algorithm's working.

**Challenge#2:** What should be the Placement of biclusters in $n \times n$ square grid: There are two principles that govern the placement of biclusters in $n \times n$ square grid:

**Principle#1:** Biclusters with high similarity should be in close proximity with each other on square grid.

**Principle#2:** Biclusters having less similarity values should be furthest to each other on square grid.

Space-filling curves [53] provides us a way to construct a curve that can cover all the points in the space while maintaining the property that close proximal points within space should be closed to each other while points having high dissimilarity should be farthest from each other. For two-dimensional space/grid like in our case this curve should cover each cell of a grid in an organized manner. Those cells which are similar should be in close proximal distance while those cells which share a good difference as compared to one another should be placed farther in the grid. There are several types of methods [54] which are used to construct space filling curves. Row filling, column filling and cantor diagonal filling methods are some of famous methods for space filling curves. Each method has its own unique properties, with usage depends on type of problem which we are dealing with. In row filling method, we traverse the matrix from left to right, top to bottom thus traversing each element line by line. In column order filling, we traverse the matrix in vertical fashion column by column. The cantor diagonal filling method traverse the matrix by its cantors. It involves traversing the matrix in diagonal manner. By accessing nearby elements diagonally, cantor diagonal space-filling traversal enhances locality and decreases memory hops. In contrast to row/column order traversals, it reduces cache misses and improves temporal consistency. It promotes a data access pattern that is more productive by balancing horizontal and vertical traversals. Thus, cantor diagonal approach has been used to fill space of two-dimensional grid of proposed work.

Fig 1 provides a comprehensive depiction of the operational dynamics involved in Cantor diagonal traversing method while filling/placing biclusters within a two-dimensional grid. In this representation, a reinforcement learning agent is initiated to explore the grid once biclusters are strategically positioned. However, the exhaustive exploration of the entire environment demands a substantial amount of time and computational resources. Consequently, intricate grid positioning techniques like Hilbert curve and Z-order have been omitted due to their complexity and potential computational overhead. The placement positions of various biclusters are detailed in section IV for each dataset, with these biclusters generated through the implementation of five distinct biclustering algorithms employed in the proposed work. The evaluation of bicluster quality is facilitated through the application of Jaccard similarity measure, and subsequently, these biclusters are mapped onto the grid for the reinforcement learning agent to traverse and navigate through the environment.

Navigating the grid efficiently is crucial for optimizing the performance of the reinforcement learning agent, and the choice of bicluster placement can significantly impact the agent's

| | 1 | 2 | 3 | 4 | 5 | 6 |
|---|---|---|---|---|---|---|
| 1 | B1 | B3 | B4 | B10 | B11 | B16 |
| 2 | B2 | B5 | B9 | B12 | B17 | B26 |
| 3 | B6 | B8 | B13 | B18 | B25 | B27 |
| 4 | B7 | B14 | B19 | B24 | B28 | B33 |
| 5 | B15 | B20 | B23 | B29 | B32 | B34 |
| 6 | B21 | B22 | B30 | B31 | B35 | B36 |

**Fig 1. Cantor-diagonal traversal of 6×6 grid.**

learning process. The utilization of diverse biclustering algorithms allows for the exploration of varied patterns and structures within the data, contributing to a more nuanced understanding of the underlying relationships. The proposed quality measures serve as benchmarks for evaluating the effectiveness of the generated biclusters, providing insights into their utility for the reinforcement learning agent. As the agent interacts with the strategically placed biclusters, this approach aims to strike a balance between computational efficiency and the ability to capture intricate patterns, fostering an adaptive and informed exploration of the grid in the context of reinforcement learning.

## E. MDP notation

RL underlines the core concept of machine learning where agent iterates over a particular environment, takes possible actions with aim to maximize cumulative reward. RL can be modeled as Markov decision process (MDP) that consists of state space covering all possibilities and actions that an agent may take. Transition possibilities that show how state changes when an agent takes an action in order to maximize cumulative reward.

For current problem, we considered state space as finite number of biclusters placed on the grid, state space $S = \{B_1, B_2, B_3, B_4 \ldots B_n\}$. At each time instant $t$ the agent recieves some representation of environment with state $S_t \in S$. For the transition, agent selects an action from the action space $A = \{Up, Left, Down, Right,\}$, $A_t \in A$. Afterward, the agent is moved to a new state $S_{t+1} \in S$ for time $t+1$. During this process, the agent will receive a reward $R_{t+1} \in R$. This process is repeated where agent will aim to select those actions which helps to maximize cumulative reward. A policy will be learned by the agent in each episode which shows sequence of actions that should be taken to maximize reward.

**Start state.**   Start state is the state within grid from which RL agent begin its movement. The work [55] uses Jaccard similarity to predict about start state. However, limitation of Jaccard is that it does not include rating values and its value may remain consistent for numerous biclusters. The work [56] used cosine similarity to predict about start state. However, cosine similarity only focuses on angle between voting vectors not on the length of voting vectors. To overcome these limitations, we have used ITR similarity as metric to predict start state for the RL agent [57]. The main reason for its usage is that it include users' features along with voting values to enable better measure to predict ratings.

**Hole state.**   When agent is traversing through the RL environment and it encounters a situation where there is no overlapping of users and items between current and next state, then this state is known as a hole state. Agent can't move forward from the current state as reward is zero due to zero overlapping.

**Reward function.**   Reward function is computed as a measure of overlapping of users in current and prospective next state.

$$R_{t+1} = \frac{|U_{st} \cap U_{st+1}|}{|U_{st} \cup U_{st+1}|} \tag{7}$$

**Goal stat.**   A goal state can be defined as a state after which no new items are added to the recommendation set of the active user while it visits different states. Moreover, goal state can be a situation where agent again moves back to start state or get off from the grid.

**Number of episodes and steps in each episode.**   The episode count is set at a constant value of 100. The number of steps within each episode is constrained by the grid size. In the case of our grid, which encompasses 36 states and features an action space with 4 available actions, the permissible number of steps is determined by multiplying the number of states by the number of actions in the action space. This computation yields a maximum of 144 steps for each episode.

**Maximizing return.**

$$G_t = R_{t+1} + \gamma R_{t+2} + \gamma^2 R_{t+3} + \gamma^3 R_{t+4} \cdots$$

$$G_t = \sum_{k=0}^{\infty} \gamma^k R_{t+k+1} \tag{8}$$

In MDP, cumulative reward is commonly referred to as the "return," which calculates the entire number of rewards an agent accrues over the course of a series of acts in an environment. The agent experiences immediate rewards for every action it does while interacting with the environment. The cumulative reward accounts for both the present benefits and the future benefits an agent anticipates from adhering to a particular policy. It essentially captures the agent's goal to maximize its long-term advantage by choosing activities that provide the most beneficial results. The discount factor, which is a parameter with the symbol "γ" (gamma) in RL, is a measure of how much an agent values future benefits relative to immediate rewards. It has a range of 0 to 1, with values closer to 1 indicating that the agent significantly prioritizes long-term rewards and closer to 0 indicating that short-term advantages are prioritized as shown in Eq (8)

Where $\gamma$ is a discount factor used to give large importance to current reward and least importance to future rewards. Last but not least, the agent's goal is to relocate to a state that offers greater reward value, which in turn increases return $G_t$. Complete working of our proposed methodology is given in Fig 2.

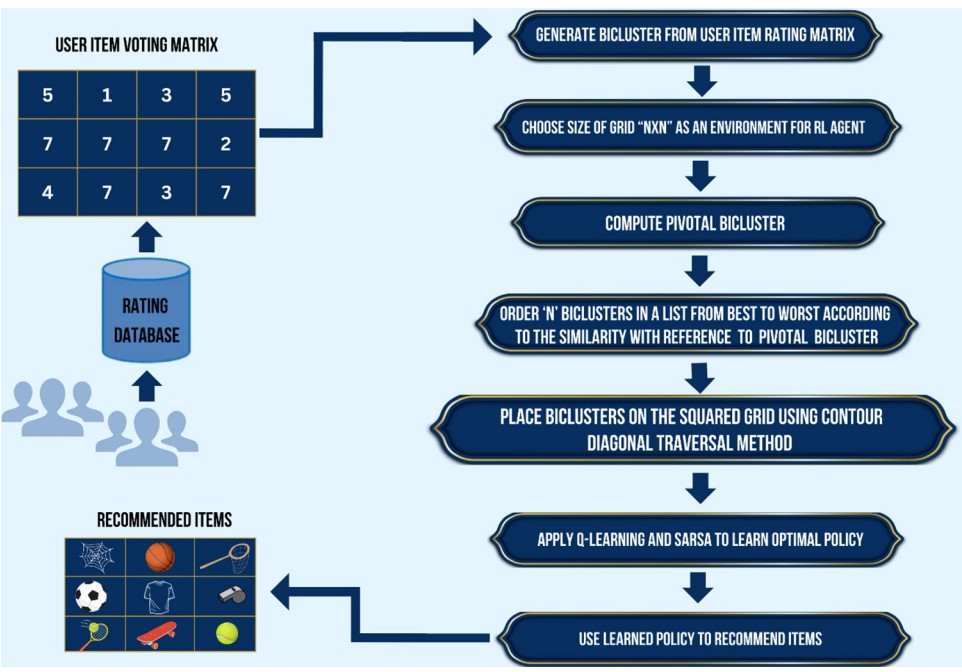

**Fig 2. Flow diagram of proposed methodology.**

**Optimal policy $\pi^*(s)$.** A policy $\pi(s)$ is a concept that maps states to actions $a \in A$ and specifies what the agent should do in each state $s \in S$. The policy can be predictable, where it explicitly translates states to particular actions, or random actions and gives a probability distribution over a set of actions for each state. Finding an optimal policy $\pi^*(s)$ by a state-action value function $Q_\pi(s, a)$, enhances the anticipated cumulative reward $G_t$ across the agent's interactions with the environment as given in Eq (9):

$$Q_\pi(s, a) = E_\pi\{G_t | s_t = s, a_t = a\}$$

$$= E_\pi\{\sum_{k=0}^{\infty} \gamma^k R_{t+k} | s_t = s, a_t = a\} \tag{9}$$

The agent initially begins exploring the environment because it has no knowledge of it. Agent begins utilizing the environment as soon as it begins to learn more about it to balance exploration and exploitation. Epsilon greedy method is used to balance exploration and exploitation. Here we are using Q-learning to learn optimal policy. Q-learning builds a Q-table containing all possible states and actions as shown in Table 3.

**Table 3. Snapshot of Q-table at some time instance for Q-learning.**

| States | Bicluster Number | Actions | | | |
|---|---|---|---|---|---|
| | | Left | Right | Up | Down |
| | B1 | 35.2577 | 35.2579 | 37.0541 | 37.1140 |
| | B2 | 35.2583 | 37.1140 | 37.0478 | 36.9598 |
| | B3 | 33.6751 | 37.0356 | 34.0091 | 35.3424 |
| | B4 | 17.7066 | 36.1686 | 25.3526 | 5.2804 |
| | ------ | 0.0000 | 14.9489 | 0.0000 | 0.0000 |
| | Bn | 0.0000 | 2.4103 | 0.0000 | 0.0000 |

In [Table 3](), if environment is deterministic, Agent will start from $B_1$ state, will take action down having highest reward 37.1140 among all other action rewards and will land to state $B_3$. In state $B_3$ agent will take action right having probability 37.0356 and will land to next state $B_5$. In this way agent will continue its move until it reaches to a goal state or to a pit/hole state. Mathematically it is given by Eq ([10]):

$$Q(s, a) = Q(s, a) + \alpha*(R + \gamma*\max(Q(s', a')) - Q(s, a)) \tag{10}$$

Where $Q(s, a)$ represents expected reward for the state $s$ while action $a$ is taken. Moreover, $R$ is actual reward with, $\gamma$ are learning and discount factor for the given Q learning problem. Moreover, the maximum expected reward is represented by $\max(Q(s', a'))$. The value of $\alpha$ varies from 1 to 0 where 0 highlights that Q value will be updated on past experience while for $\alpha = 1$ Q value will be updated on current experience. Moreover, discount factor $\gamma$ will be used to select corresponding contributions of future and current rewards.

Another technique that is used to learn optimal policy is SARSA. SARSA (State-Action-Reward-State-Action) is a reinforcement learning algorithm that operates on the on-policy approach, estimating Q-values based on the current policy. It also uses same Q table to learn optimal policy regarding RL environment as of Q-learning. However, there are some key differences. SARSA follows an on-policy approach, meaning it estimates Q-values based on the current policy and takes the actual next action in the environment. In contrast, Q-learning is an off-policy algorithm that estimates Q-values regardless of the current policy and generally selects the action with the highest Q-value. The update rule for Q-learning is based on the maximum Q-value of the next state-action pair, emphasizing a more exploratory strategy. While SARSA is often considered more conservative and safer, Q-learning tends to be more aggressive in its pursuit of optimal policies, which can lead to more efficient learning in certain scenarios. Mathematically SARSA is given by Eq ([11]):

$$Q(s, a) = Q(s, a) + \alpha*(R + \gamma*Q(s\prime, a\prime)) - Q(s, a)) \tag{11}$$

A popular strategy in reinforcement learning is to balance exploitation and exploration in an agent's decision-making process. We are using epsilon-greedy method to balance exploitation and exploration. This approach gives the agent the option of either exploring new options or taking advantage of the best option that is currently known. This mechanism ensures selection of best action with probability $1-\epsilon$ and choosing random action with probability $\epsilon$. The parameter $\epsilon$ is frequently decayed over time in epsilon-greedy method to include relative decay. The relative decay makes sure that as the agent gets more knowledge about environment, it starts to exploit more than to explore. In proposed work, decay is represented as an exponential decay function whose value decreases with time as shown in Eq ([12]).

$$\epsilon_t = \epsilon_0 \cdot e^{-\alpha t} \tag{12}$$

where:

- $\epsilon_t$ is the exploration probability at time $t$.

- $\epsilon_0$ is the initial exploration probability.

- $\alpha$ is the decay rate.

- $t$ is the time or the number of steps taken.

The value of epsilon will be used in the epsilon-greedy method. Thus agent selects the optimal action with probability $1-\epsilon$ and explores with probability $\epsilon$. The action is chosen based on the Q-value function, where with probability $1-\epsilon$, the agent selects the action with the highest

Q-value, and with probability $\epsilon$, a random action is chosen. The epsilon-greedy strategy is expressed as in Eq (13). Fig 2 shows overall workflow of the proposed work.

$$Action\ a(t) = \begin{cases} \max Q_{t(a)} \text{with probability } 1 - \varepsilon \\ any\ action\ with\ probability\ \varepsilon \end{cases} \tag{13}$$

## F. Rating prediction

In order to recommend items to a target user, learned optimal policy $\pi^*(s)$ is used. This policy is composed of actions that leads us to a goal state. Policy actions are chosen based on highest $Q(s, a)$ at a particular state $s$.

$$\pi^*(s) = \begin{cases} a(t) for\ state\ s\ where\ argmax_{a \in A} Q(s, a) \\ random\ action\ a(t) from\ A \end{cases} \tag{14}$$

Each action leads to a state which represents a bicluster containing a set of items. Thus recommended items $\hat{I}$ will be a set of items in the item set ($BI$) of these biclusters.

$$Recommended\ items\ I_P = \{BI_{a(t)} \cup BI_{a(t+1)} \cup BI_{a(t+2)} \ldots \cup BI_{a(t+n)}\} \tag{15}$$

Prediction of rating for RL based recommender system has been unprecedented so far. The main reason for this is on contrary to existing machine learning techniques, RL agent does not construct a mathematical model/function that can predict rating based on certain optimization using a particular loss function. It generates recommendation by just traversing on RL environment according to certain reward function. It does not involve learning of latent features to capture user's preferences and item features. The reward function in general guides agent for future action while giving no support to predict rating for recommended items. It is beneficial to predict ratings in a recommendation system from a group of users who are similar because it makes use of the knowledge of others and enables more specialized and precise recommendations. In this work, we are proposing column average of each individual item in a bicluster as a predicted rating value for recommended item. Let's assume we have a bicluster having $m$ number of rows and $n$ number of columns. Mathematically expressing, column average in a bicluster is the sum of all the values present in that particular bicluster column divided by number of non-zero entries in that column. Let $B(i, j)$ represent the element at row $i$ and column $j$ of bicluster $B$. Let $N_j$ represent the number of non-zero entries in column $j$.

The column average $C_j$ for column $j$ can be expressed mathematically as in Eq (16):

$$C_j = \sum B(i, j)/N_j \tag{16}$$

Where:

- $B(i, j)$ is the value of the element in row $i$ and column $j$.

- $i$ iterates over the rows of $B(i, j)$.

- $N_j$ is the number of non-zero entries in column $j$.

- $m$ represents number of rows in a particular bicluster

In this way column average is calculated for each column which represents a single item across a particular bicluster. However, there may be a case that there exists many of the overlapping elements within various biclusters. For any two biclusters having some common items

between them, rating of common items will be updated by taking mean of average rating for all common items in each of the bicluster. Let's say we have two Bicluster B1 and B2 with common columns or Items $I_1, I_2. . . .I_n$ and we want to calculate the average rating of the common items for each of the two biclusters. It can be expressed mathematically as in Eq (17):

$$P_{r(i)} = \sum_{i=I_1}^{I_n} (C_{i(1)} + C_{i(2)})/2 \tag{17}$$

Where:

- $I_1, I_2, . . . ..., I_n$ represents common items among two biclusters

- $C_{i(1)}$ is average column value of $i^{th}$ common item in bicluster 1

- $C_{i(2)}$ is average column value of $i^{th}$ common item in bicluster 2

- $P_{r(i)}$ is aggregated/predicted rating of common items between two biclusters B1 and bicluster B2

In this way, average rating of overlapping items across various biclusters is updated when same items are encountered. Moreover, these ratings are constantly updated during the process and final items are recommended with updated ratings containing aggregated rating of overlapping items. This approach allows integration of the tastes and actions of a group of users who are similar to the target user in terms of appearance and interests. Every user in this group provides their own ratings and circumstances, which together create a variety of perspectives.

**Example scenario.** To further illustrate the working mechanism of proposed algorithm, consider a sample grid as shown in Fig 3(A). It consists of nine biclusters. Let assume start state for agent is B1 which is calculated using triangular similarity. The agent will traverse through the grid using four possible actions i.e. up, left, down and right. The numeric values 0,1,2 and 3 are assigned to these actions respectively. Let assume the learned policy is

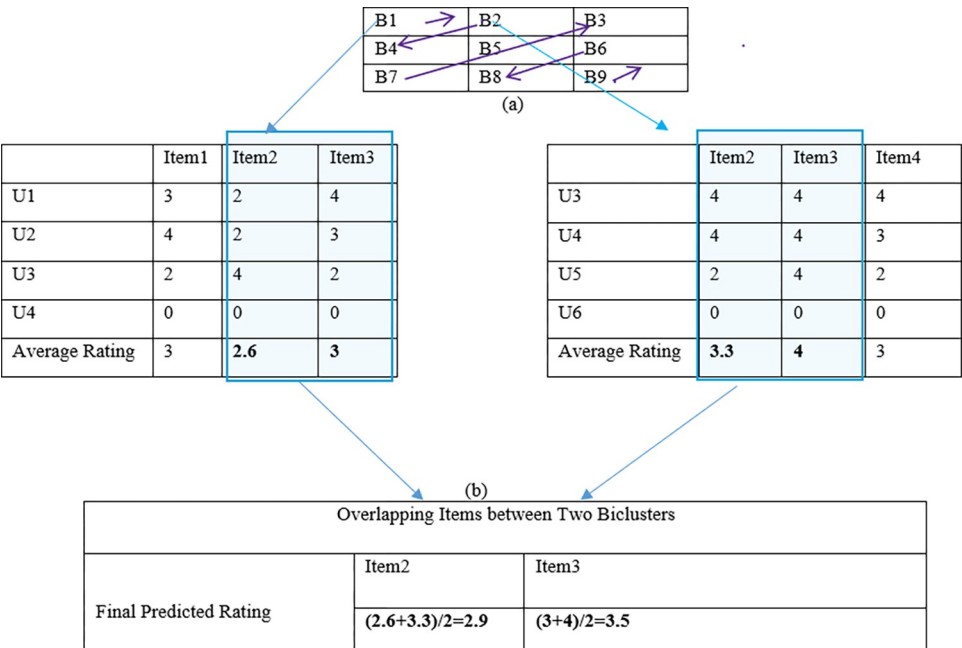

**Fig 3. Example scenario to illustrate how rating is predicted by proposed algorithm.**

[3,2,0,0,0] so according to policy the agent will move one step right, one step down and then three steps traversing in upward direction thus moving from B1 to B2, B4 B7, B5 and B3. Furthermore, it shows two of sample biclusters from the grid B1 and B2. It can be seen from these two biclusters that U3 and U4 are overlapping users and Item2 and Item3 are overlapping items. The agent moves in the grid in the contor diagnol fashion using learned policy which was extracted by considering the maximum value of reward. Moreover, to predict rating, average rating of each item in individual bicluster is calculated by dividing sum of rating of that item with number of users who have rated that item. In case a user has not rated that item, the rating for that user will be zero and it will not be used in calculation of average rating for that item. Furthermore, if there are overlapping items among various biclusters then final ratings of these overlapping items will be calculated by taking mean of their average ratings across various biclusters as shown in Fig 3(B).

## IV. Experimentation procedure

In this section we will give detailed description of datasets we used for experimentation, methods used for comparison and metrics used for evaluation purpose. We have used MovieLens ML-100K [58], ML Latest-Small [59] and FilmTrust [60] datasets for experimentation purpose. The ML-100K dataset comprises of 100,000 ratings given by 943 users on 1,682 distinct movies. ML-latest small dataset consists of 5-star ratings with over 9,742 movies, it has 100,836 ratings and 3,683 tag applications.

FilmTrust dataset comprises of 35,497 ratings given by 1,508 users on 2,071 distinct movies. Due to relatively large size of datasets, the input space becomes too large for RL agent to work upon if we apply RL directly to original user-item ratings matrix of these datasets. For this reason, biclustering technique is applied on the given dataset prior to feeding it to the RL agent so that less time and cost will be taken by RL agent to generate recommendations. Table 4 shows representations of various grid positions of generated biclusters after applying various biclustering algorithms. These varying grid positions show that Jaccard similarity measure have placed biclusters in different positions which remain specific to dataset and particular algorithm that is being used to generate biclusters. Pycharm community edition is used as IDE. Code is executed on a core-i5 Dell laptop with 8-GB RAM. For squared grid environment creation OpenAI gym is used.

The Table 4 highlights a detailed portrayal of the configurations of 36 biclusters within a 6×6 grid, each derived from the application of different biclustering algorithms to distinct datasets. The dynamic shifts in the positions of these biclusters underscore the nuanced adaptability of different biclustering algorithms. This flexibility is particularly noteworthy as it intelligently situates biclusters in varying positions, tailoring its outcomes to the unique characteristics inherent in each specific dataset. This dataset-specific positioning demonstrates the ability of biclustering algorithm to discern and respond to the intricacies and patterns embedded in different datasets. These varying grid positions will become an environment for RL agent to explore and recommend items to active user.

## V. Results and discussion

We compared our proposed method with a variety of RL and CF based methods. Detail of these methods is given below

### RLRS1 [56]

This method applied RL on biclusters placed on an $n \times n$ square grid. Biclusters are generated using only one biclustering algorithm which is bimax. They used a filtering mechanism to filter out good biclusters using MSR quality measure, leaving out bad quality biclusters, which are

**Table 4. Grid positions of biclusters across three datasets for used biclustering algorithms.**

| Algorithm | ML-100K | | | | | | ML-Small | | | | | | Film Trust | | | | | |
|---|---|---|---|---|---|---|---|---|---|---|---|---|---|---|---|---|---|---|
| Bibit | 1 | 14 | 17 | 26 | 20 | 24 | 1 | 17 | 13 | 35 | 2 | 36 | 1 | 18 | 12 | 35 | 25 | 8 |
| | 21 | 13 | 5 | 19 | 22 | 32 | 11 | 31 | 20 | 6 | 19 | 25 | 2 | 20 | 28 | 27 | 4 | 11 |
| | 18 | 4 | 23 | 12 | 7 | 8 | 4 | 27 | 24 | 33 | 23 | 9 | 23 | 10 | 13 | 19 | 32 | 31 |
| | 25 | 9 | 2 | 33 | 3 | 15 | 3 | 21 | 29 | 30 | 15 | 18 | 26 | 24 | 6 | 16 | 36 | 3 |
| | 11 | 28 | 6 | 36 | 34 | 16 | 10 | 35 | 7 | 32 | 28 | 26 | 17 | 33 | 14 | 15 | 22 | 5 |
| | 35 | 29 | 9 | 30 | 29 | 31 | 22 | 14 | 12 | 16 | 5 | 8 | 7 | 21 | 9 | 29 | 34 | 30 |
| BiMax | 1 | 11 | 30 | 10 | 12 | 24 | 1 | 4 | 9 | 14 | 15 | 21 | 1 | 7 | 8 | 14 | 15 | 4 |
| | 4 | 5 | 9 | 13 | 27 | 20 | 10 | 11 | 8 | 16 | 22 | 34 | 5 | 9 | 13 | 16 | 5 | 26 |
| | 6 | 8 | 15 | 32 | 19 | 21 | 12 | 6 | 17 | 23 | 32 | 35 | 10 | 12 | 17 | 2 | 24 | 29 |
| | 7 | 16 | 33 | 18 | 13 | 29 | 13 | 18 | 24 | 30 | 36 | 5 | 11 | 18 | 19 | 23 | 30 | 27 |
| | 17 | 34 | 14 | 23 | 28 | 31 | 19 | 25 | 29 | 31 | 33 | 2 | 3 | 32 | 22 | 31 | 35 | 28 |
| | 2 | 3 | 25 | 26 | 36 | 35 | 27 | 28 | 20 | 26 | 3 | 4 | 20 | 21 | 33 | 34 | 25 | 36 |
| Iterative Clustering Algorithm | 1 | 14 | 21 | 11 | 2 | 27 | 1 | 30 | 24 | 6 | 36 | 4 | 1 | 14 | 21 | 16 | 20 | 7 |
| | 7 | 35 | 9 | 24 | 36 | 11 | 14 | 28 | 34 | 17 | 5 | 31 | 9 | 6 | 19 | 22 | 30 | 3 |
| | 9 | 31 | 36 | 16 | 3 | 18 | 8 | 32 | 9 | 10 | 12 | 16 | 17 | 10 | 24 | 8 | 15 | 5 |
| | 14 | 19 | 28 | 12 | 15 | 5 | 20 | 15 | 19 | 22 | 23 | 27 | 29 | 26 | 4 | 18 | 35 | 36 |
| | 29 | 20 | 4 | 25 | 22 | 30 | 26 | 11 | 2 | 3 | 33 | 29 | 2 | 12 | 28 | 23 | 27 | 25 |
| | 33 | 17 | 26 | 23 | 10 | 32 | 13 | 21 | 25 | 7 | 35 | 18 | 31 | 11 | 13 | 34 | 32 | 33 |
| Bipartite | 1 | 2 | 3 | 9 | 10 | 15 | 1 | 33 | 2 | 9 | 9 | 14 | 1 | 2 | 3 | 9 | 10 | 15 |
| | 25 | 4 | 9 | 11 | 16 | 26 | 34 | 3 | 7 | 10 | 15 | 24 | 35 | 4 | 8 | 11 | 16 | 25 |
| | 5 | 7 | 12 | 17 | 24 | 27 | 4 | 6 | 11 | 16 | 23 | 25 | 5 | 7 | 12 | 17 | 24 | 26 |
| | 6 | 13 | 18 | 23 | 28 | 33 | 5 | 12 | 17 | 22 | 26 | 31 | 6 | 13 | 18 | 23 | 27 | 32 |
| | 14 | 19 | 22 | 29 | 32 | 34 | 13 | 18 | 21 | 27 | 30 | 32 | 14 | 19 | 22 | 28 | 31 | 33 |
| | 20 | 21 | 30 | 31 | 35 | 36 | 19 | 20 | 28 | 29 | 35 | 36 | 20 | 21 | 29 | 30 | 34 | 36 |
| Least Average Value | 1 | 33 | 13 | 14 | 16 | 21 | 1 | 31 | 15 | 10 | 19 | 25 | 1 | 23 | 9 | 24 | 13 | 10 |
| | 5 | 23 | 11 | 9 | 36 | 2 | 32 | 2 | 8 | 32 | 29 | 18 | 14 | 2 | 7 | 3 | 22 | 36 |
| | 34 | 15 | 22 | 33 | 18 | 25 | 33 | 35 | 7 | 30 | 11 | 9 | 15 | 16 | 28 | 6 | 25 | 20 |
| | 10 | 28 | 4 | 6 | 27 | 30 | 4 | 3 | 22 | 24 | 20 | 5 | 33 | 30 | 5 | 29 | 26 | 35 |
| | 35 | 3 | 17 | 26 | 20 | 31 | 34 | 23 | 26 | 12 | 28 | 36 | 11 | 4 | 31 | 8 | 17 | 19 |
| | 19 | 23 | 8 | 29 | 12 | 32 | 27 | 6 | 13 | 21 | 16 | 17 | 21 | 27 | 32 | 34 | 12 | 18 |

not placed on the squared grid ultimately. Leaving bad quality biclusters result in information loss and effects recommendation accuracy. Start state is determined using Cosine similarity. No mechanism to predict ratings for recommended items is presented.

## RLRS2 [34]

This method applied RL on biclusters placed on an $n \times n$ square grid. Biclusters are generated using two biclustering algorithms namely bibit and bimax. Reward function is based on overlapping of users and items. The quality measure used in this work to measure quality of biclusters and reward function to traverse through the grid do not synchronize well. Moreover, there was no mechanism proposed to predict rating of recommended items. Start state is determined using triangular similarity.

## IPWR (Pure CF method) [61]

In this technique, the recommendation of items to users involves computing similarity using an enhanced PCC measure that incorporates both user and item averages. The enhanced PCC is subsequently merged with users' rating preference behavior (RPB).

### ITR (Pure CF method) [57]

In this approach, the similarity between two users is determined by representing their rating vectors as a triangle. Calculated triangle similarity is then integrated with the users' rating preferences (URP).

### S4 (Partitional clustering method) [62]

This study proposed eighteen distinct approaches for determining the initial centroids of clusters by manipulating the inherent correlation structure of the data. From these options, we specifically opted for method 4, referred to as S4 in the research paper. The S4 method selects $k$ users as initial centroids, focusing on their density within a multidimensional sphere.

### SPOP (Partitional Clustering method) [63]

This method segregates users into different hyperspheres by assessing their distance from each hypersphere's center, using the Pearson correlation with default votes. However, regarding default votes for non-rated items is merely an approximation and is not ideally suitable, as users may hesitate to vote for items with approximate vote values, impacting the method's reliability.

### A. Evaluation metrics

Apart from precision and recall, some new evaluation metrics have gained popularity in recent times [64, 65] by considering the dynamic nature of user's interest. It is crucial to assess recommendation algorithm in terms of some new measures like personalization, intra-list similarity, and novelty as this gives us broader view to assess recommendation quality rather than just sticking to traditional evaluation metrics like precision and recall. This will ensure that recommendation methods not only deliver relevant content but also offer diverse, personalized, and novel experiences, ultimately leading to more effective and user-centric recommendation services. In following section, we will define these evaluation metrics in detail

**Personalization.** Measured as Latency $H(L)$. It considers the uniqueness in different users' recommendation lists—that is, inter-user diversity. Given two users $i$ and $j$, the difference between their recommendation lists can be measured by the inter-list distance as shown in Eq (18).

$$H_{ij}(L) = 1 - \frac{Q_{ij}(L)}{(L)} \tag{18}$$

where $Q_{ij}(L)$ is the number of common items in the top L places of both users recommended lists: identical lists thus have $H_{ij}(L) = 0$ whereas completely different lists have $H_{ij}(L) = 1$. Thus, value of personalization varies from 0 to 1 where 0 means no personalization and 1 means maximum personalization.

**Novelty.** It measures the unexpectedness of an item relative to its global popularity. Given an item α, the chance that a randomly selected user has collected α is given by $k_u$/u given as in Eq (19).

$$I_a = LOG2(U/Ka) \tag{19}$$

Here $a$ is a specific item, u is total number of users, $k_u$ is the number of users who have interacted with at least one item in the system and $I_a$ is novelty of item $a$. Higher positive value of $I_a$ shows high novelty. This logarithmic measure encourages the recommendation system to

prioritize diverse and less-interacted-with items, enhancing the user experience with a variety of suggestions.

**Intra list similarity.** It is the average cosine similarity of all items in a recommendation list. This calculation uses features of the recommended items (such as movie genre) to calculate the similarity. If a recommender system is recommending lists of very similar items to single users (for example, a user receives only recommendations of romance movies), then the intra-list similarity will be high. A low intra list similarity indicates more diverse recommendation. It values ranges from 0 to 1 where 1 means high similarity and value close to 0 indicate highly dissimilar and diverse recommendations. This metric can help to understand how much diverse recommendation is being generated by RL agent.

**Mean absolute error (MAE) and root mean square error (RMSE).** MAE is a measure of the average squared differences between the predicted ratings (or scores) and the actual ratings provided by the user as shown in Eq (20).

$$MAE = \frac{1}{n} \sum_{i=1}^{n} \left| P_{r(i)} - A_{r(i)} \right| \tag{20}$$

- $n$ is the total number of ratings or recommendations.

- $P_{r(i)}$ is the predicted rating or score for item $i$.

- $A_{r(i)}$ is the actual rating provided by the user for item $i$

RMSE is the square root of the MSE and provides a more interpretable metric in the same units as the original ratings as shown in Eq (21).

$$RMSE = \sqrt{\frac{1}{n} \sum_{i=1}^{n} \left( P_{r(i)} - A_{r(i)} \right)^2} \tag{21}$$

**Precision.** It is the ratio of correctly recommended items to the total number of recommended items. For testing purpose, we kept values of $|I_P|$ to 10 and 20 during experimentation as shown Eq (22).

$$Precision = \frac{|I_T \cap I_P|}{|I_P|} \tag{22}$$

**Recall.** It is ratio of correctly recommended items to the total number of test items as shown Eq (23).

$$Recall = \frac{|I_T \cap I_P|}{|I_T|} \tag{23}$$

**Fmeasure.** F-measure in a recommender system is a metric that combines precision and recall, providing a balanced assessment of the system's ability to accurately recommend relevant items as shown Eq (24).

$$Fmeasure = 2 \times \frac{Precision \times Recall}{Precision + Recall} \tag{24}$$

**Item coverage.** Refers to the proportion of items in the test set that the system is able to recommend to users, as shown Eq (25).

$$Item\ Coverage = \frac{|I_T|}{|I_P|} \times 100 \tag{25}$$

## B. Learning of an agent

Figs 4–6 underlines the environment learning of RL agent across various parameters. From Fig 4(A), it can be observed that return is as low as 20 in few episodes. This low value indicates that for a particular episode, there are very few overlapping users and items across various biclusters. Furthermore, a high return of up to 80 can be seen in starting episodes, which shows that during these episodes, there are high number of overlapping users across various biclusters. The average reward observed for ML-100K dataset in Fig 4(A) is round about 30. As in Fig 5(A), return on ML-latest small dataset ranges from as low as 15 to maximum of 65. These values highlight that as compared to ML-100K dataset, there are relatively less number of overlapping users and items in biclusters. The average reward observed for ML-latest small dataset is round about 30. For FilmTrust dataset Fig 6(A), return ranges from 40 to 120. This underlines that among these datasets, FilmTrust dataset has relatively high number of biclusters with high frequency of overlapping users and items. The average reward observed for FilmTrust dataset is round about 45.

Fig 4(B) shows that steps taken by agent while exploring squared grid for ML-100K dataset, ranges from as low as 20 to maximum of 110. Each step corresponds to exploration of one bicluster. It can be seen from the figure that at the start, agent takes a lot of steps while exploring environment as he has no or very little knowledge to exploit. Afterward, with each passing episode, number of steps taken by agent during an episode reduces considerably which shows that agent is learning with time. These steps for ML-latest small dataset ranges from 20 to 120 per episode as shown in Fig 5(B). For FilmTrust dataset, these aforementioned steps vary from 25 to 100 as shown in Fig 6(B). It is clear from examining all three datasets that the agent needs less steps to find an objective with every episode that goes by. This pattern shows that the agent is becoming more efficient with time, which implies that it is picking up new skills via its contacts with the outside world. In particular, it looks that the agent is using the information that it has already learned rather than starting over every time. We used the Epsilon-greedy technique in an attempt to strike an acceptable equilibrium between exploration and exploitation.

The agent's comprehension of the environment shows a discernible increasing trend with the number of episodes, as seen in Figs 4(C), 5(C), and 6(C). This is a sign of the agent's increasing proficiency over time. The epsilon value is initially set at 0.9, indicating demonstrating a stronger focus on exploration, in which the agent experiments with different activities to find the most successful ones. The epsilon value falls with increasing episode count, ultimately reaching a value of 0.1 with a decay rate of 0.099. The gradual decrease in epsilon denotes the shift from discovery to utilization. When the agent is first starting out and has little experience, the high epsilon number makes sure that it investigates a lot of different options in the environment. As the agent gains more experience and improves its comprehension, the epsilon value gradually drops. This makes it possible for the agent to use its acquired tactics more successfully, requiring fewer actions to accomplish its objectives. The diminishing curve seen in the epsilon value represents how exploration is being shifted toward exploitation as agents becomes more knowledgeable.

## C. Result analysis on ML-100K dataset

Here we performed experiments to measure the impact of five different Biclustering algorithms on ML-100K dataset. The quality of recommendations generated by the proposed

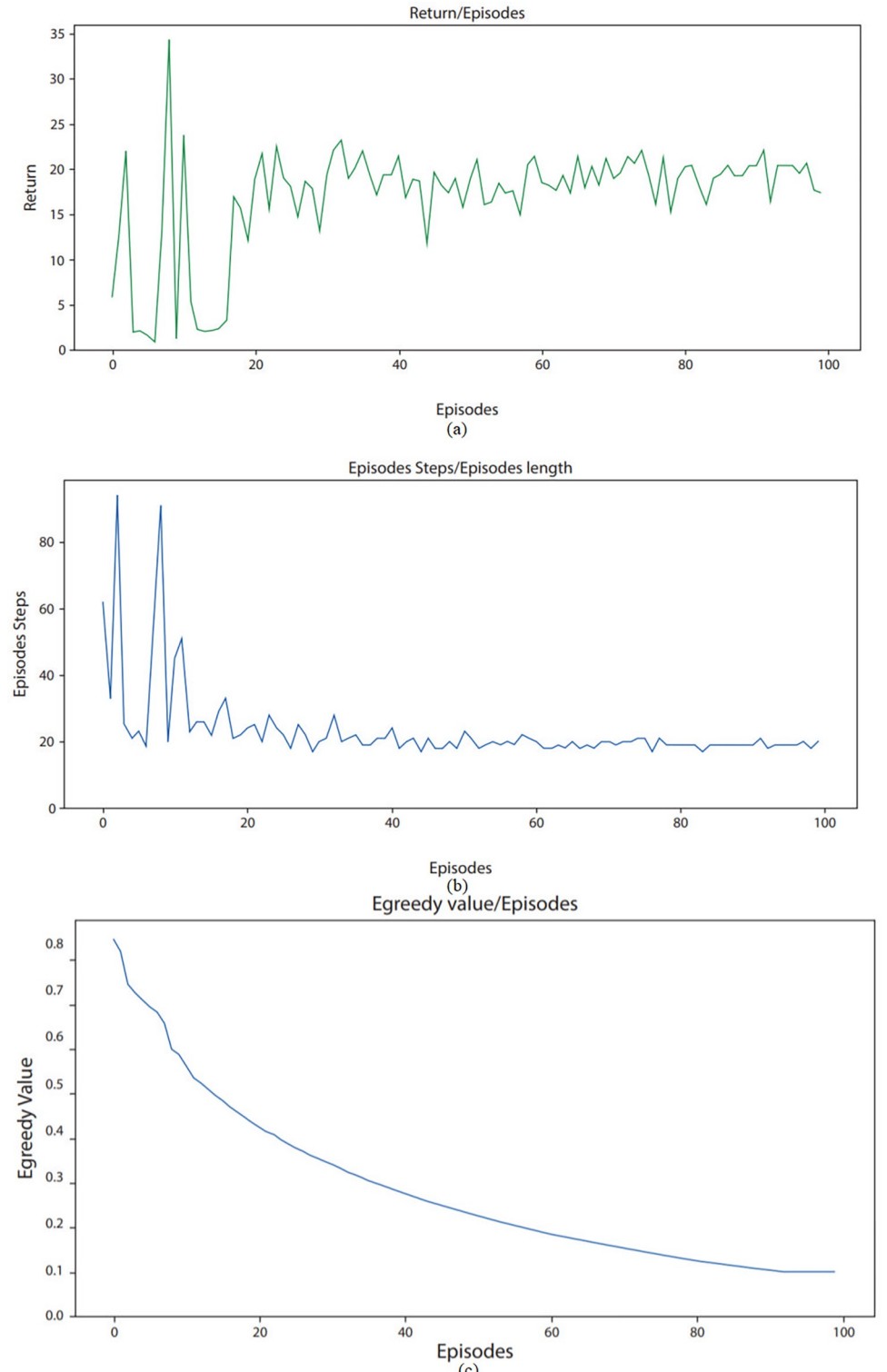

**Fig 4. Agent learning of ML100K dataset across various parameters.**

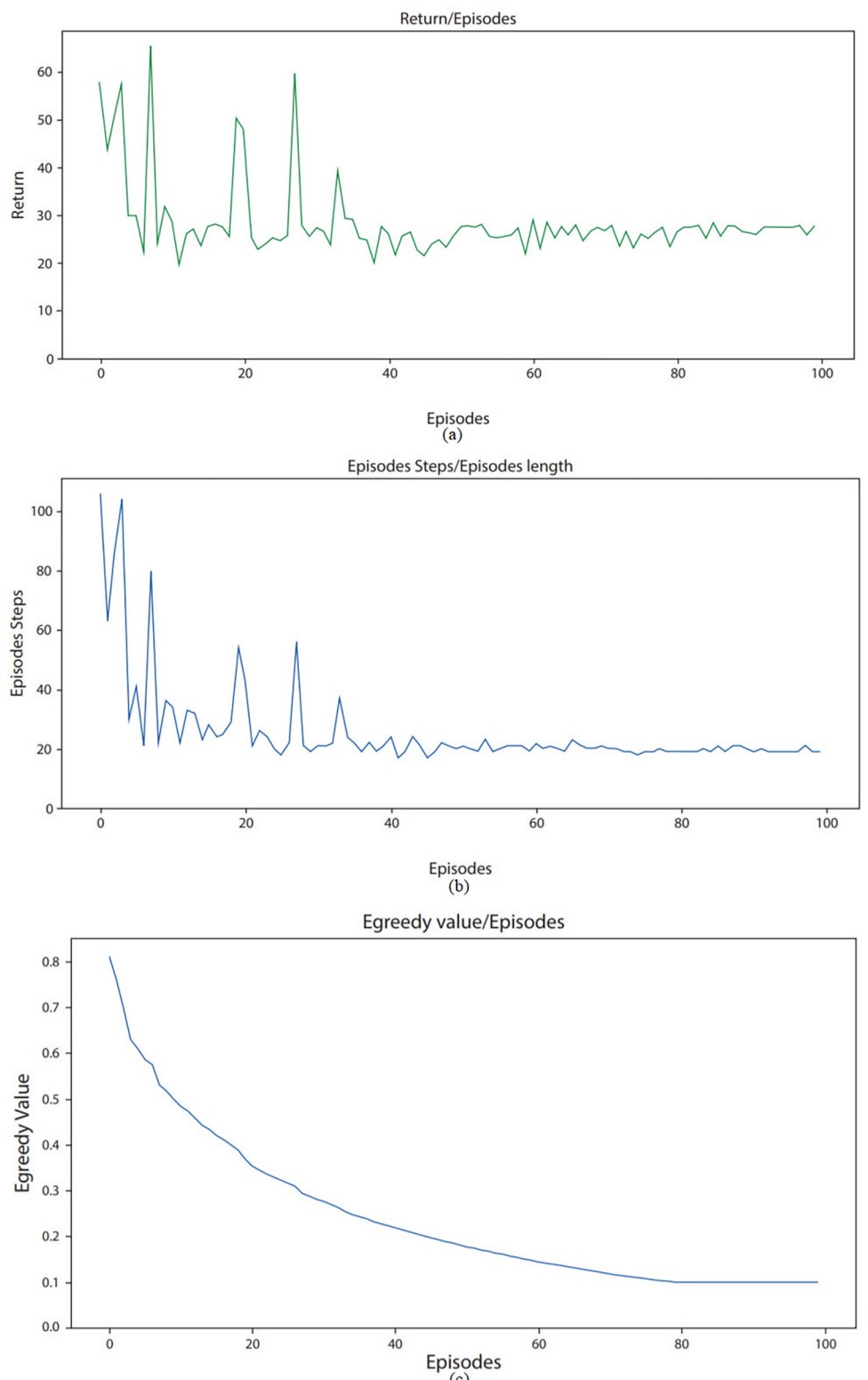

**Fig 5. Agent learning of ML latest-small dataset across various parameters.**

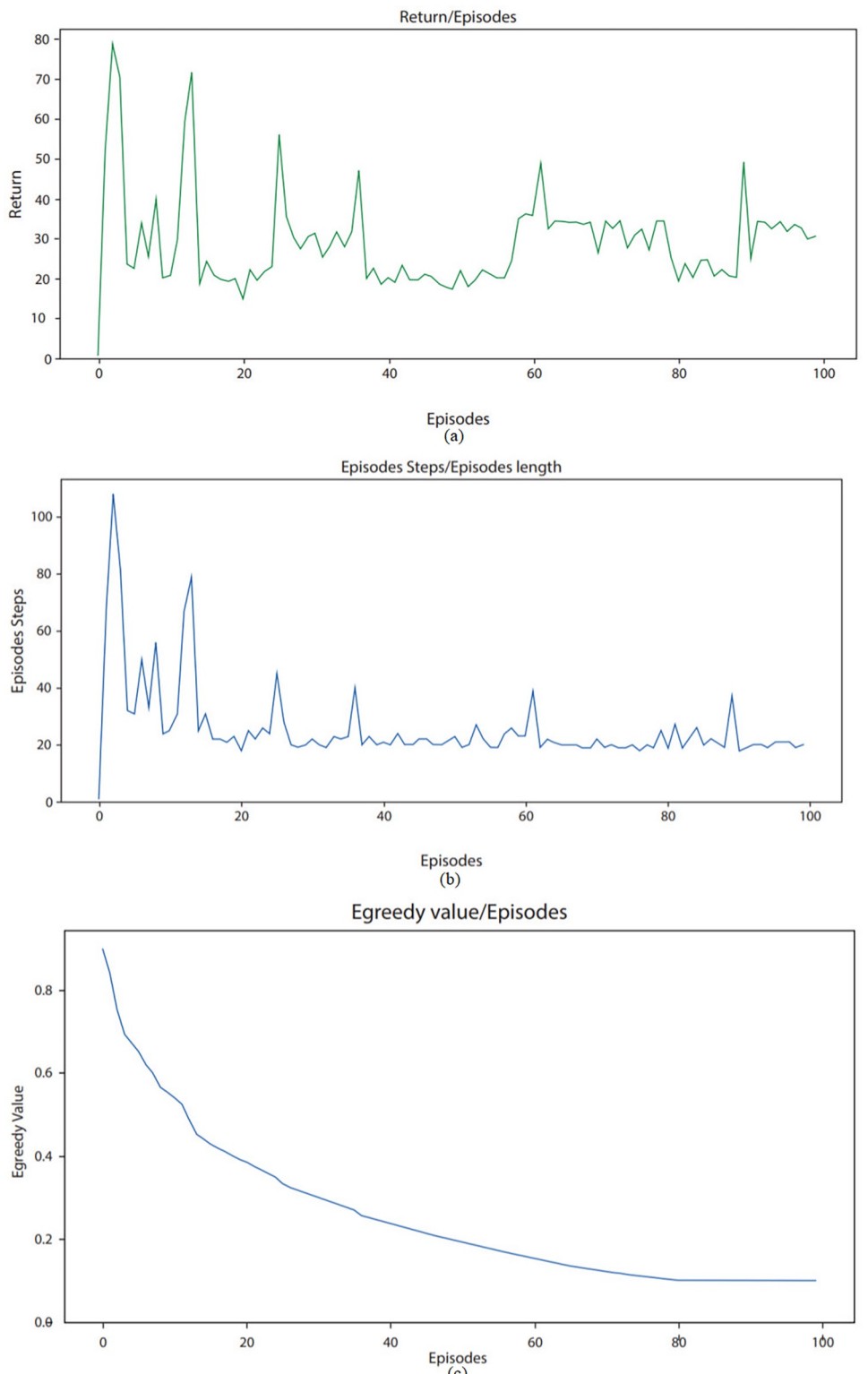

**Fig 6. Agent learning of FilmTrust dataset across various parameters.**

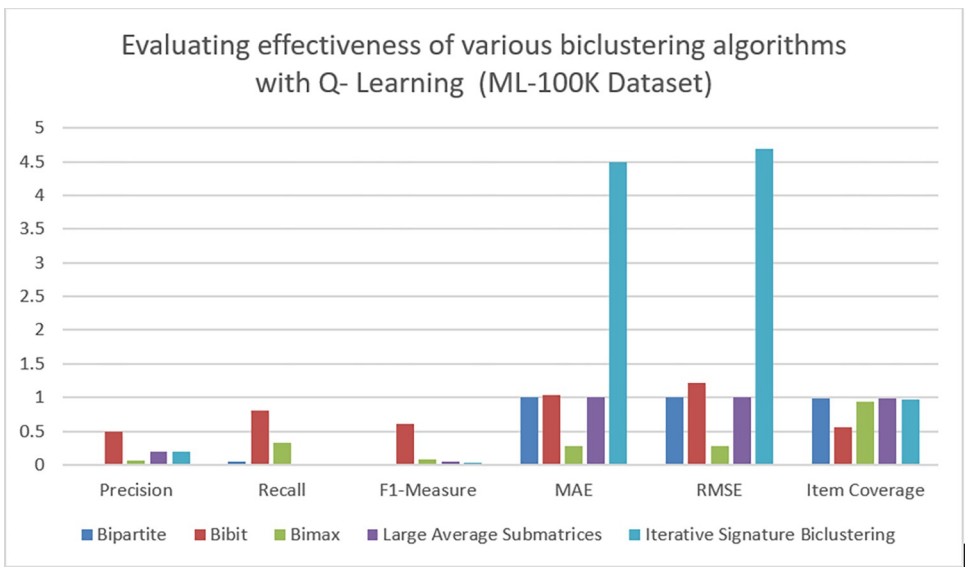

**Fig 7. Performance of various biclustering algorithms on ML-100K dataset with Q- learning.**

algorithm is evaluated in term of evaluation metrics given in section V(A). Moreover, the quality of recommendations is evaluated by generating recommendation of top items for each algorithm. Figs 7 and 8 show values of these evaluation metrics while applying Q-learning and SARSA on the given datasets. Owing to the dynamic nature of reinforcement learning based recommender systems, some specific evaluation metrices are used to evaluate diversity of generated recommendations. The enhanced diversity enables recommender system to increase visibility of less popular or niche items. Furthermore, user specific personalized recommendations are also another aspect that should be evaluated in context of RL based recommender systems. Considering above facts, personalization, intra list similarity and novelty are used as an additional evaluation metrics to measure quality of recommendation effectively. Tables 5

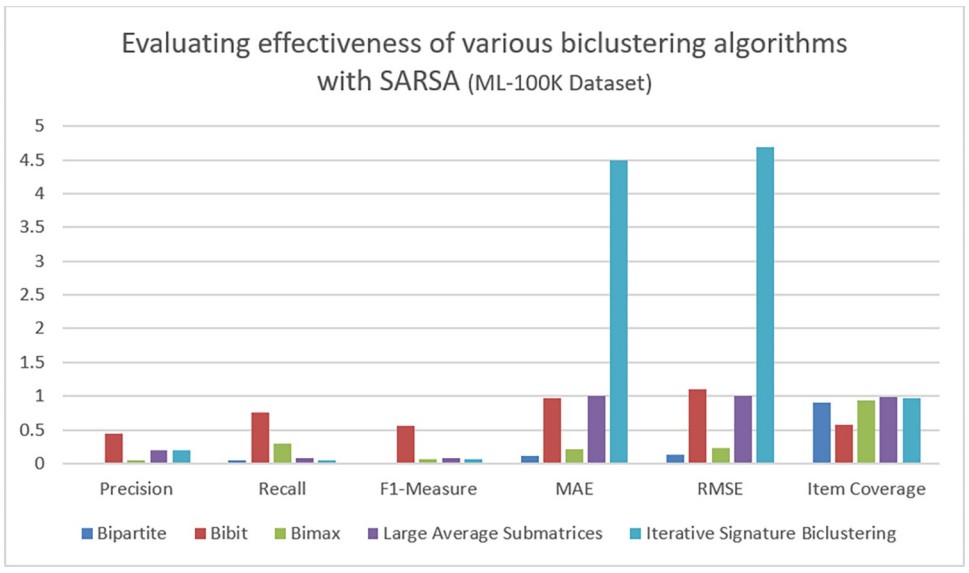

**Fig 8. Performance of various biclustering algorithms on ML-100K dataset with SARSA.**

**Table 5. Evaluation results obtained with Q-learning on ML-100K.**

| Algorithms | Personalization | Intra List Similarity | Novelty |
|---|---|---|---|
| Bipartite | 0.3 | 0.19 | 4.0 |
| Bibit | 0.6 | 0.29 | 3.5 |
| BiMax | 0.1 | 0.17 | 2.3 |
| Large Average Submatrices | 0 | 0.4 | 4.2 |
| Iterative Signature Biclustering | 0 | 0.5 | 3 |

and 6 show the values of these evaluation metrics which are highlighted as in separate tables. After analyzing results, it can be seen that Bibit biclustering algorithm has generated very personalized recommendations, when applying Q-learning on the given dataset. The high score of precision, recall, F1 measure, personalization and novelty indicates that Bibit algorithm has been able to generate recommendations that are efficient, accurate, relevant and more personalized according to active users. The low value of intra list similarity indicates that generated recommendations are diverse in nature. Apart from Bibit, no other biclustering algorithm is able to produce efficient recommendations. It has to be noted that while evaluating effectiveness of a required algorithm, the overall aggregated score of all evaluation metrics is kept as a benchmark. This enables us to understand the effectiveness of proposed algorithm across multiple dimensions. For the calculation of MAE and RMSE, proposed work represents a novel method for ratings' prediction in recommender systems based on reinforcement learning. This type of ratings' prediction was missing in existing work for RL based recommender systems. However, via experimentation it was observed that although our approach performs well enough to predict scores for a small number of objects, its effectiveness decreases as we apply it to a larger number of items. This result emphasizes the necessity of additional study to improve our method's scalability. However, our approach establishes a vital framework for future research to predict ratings and enhance the potential of recommender systems based in reinforcement learning.

## D. Result analysis on ML-latest-small dataset

We performed experiments to measure the impact of five different Biclustering algorithms on ML-Small dataset. This dataset suffers from issues like limited sample size, sparsity and biased representation. For that very reason, it's usage is very limited in research. However, due to its limited size and low cost of computation we have used it in experimentation. Moreover, we are also interested in analyzing efficiency of proposed algorithm on sparse dataset. Figs 9 and 10 show values of various evaluation metrics while applying q-learning and SARSA to the given dataset. Tables 7 and 8 show result of evaluation metrics in context of diversity for q-learning and SARSA. After analyzing results, it can be seen that BiMax biclustering algorithm has generated efficient recommendations. Moreover, the score of all other evaluation metrics

**Table 6. Evaluation results obtained with SARSA on ML-100K.**

| Top n items | Personalization | Intra List Similarity | Novelty |
|---|---|---|---|
| Bipartite | 1.0 | 0.17 | 4.3 |
| Bibit | 0 | 0.34 | 2.9 |
| BiMax | 0.4 | 0.21 | 2.2 |
| Large Average Submatrices | 0 | 0.4 | 4.2 |
| Iterative Signature Biclustering | 0 | 0.5 | 3 |

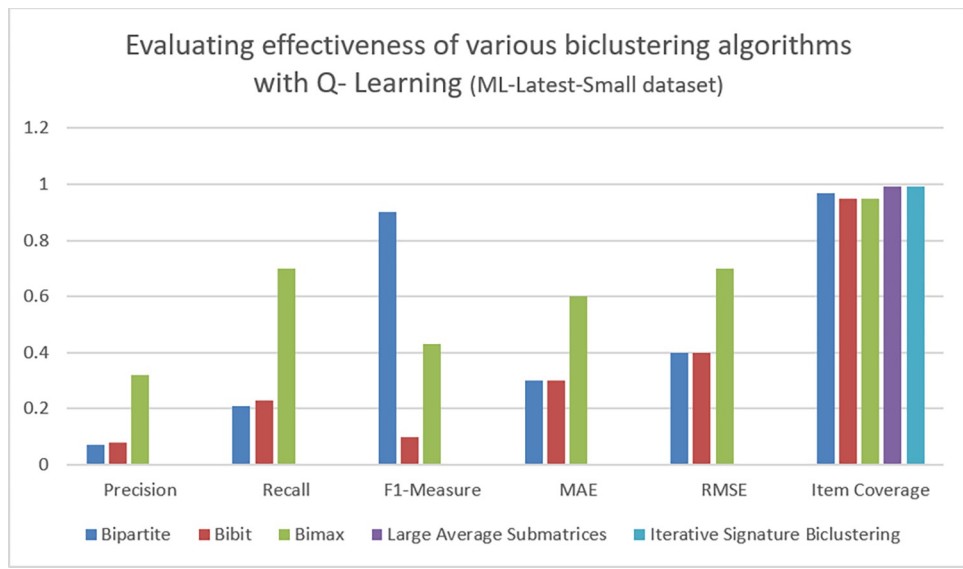

**Fig 9. Performance of various biclustering algorithms on ML-latest-small dataset with Q-learning.**

found to be good with regard to BiMax. The results obtained from BiMax biclustering algorithm highlight that generated recommendations are quite accurate, relevant, diverse and have relatively low value of MAE and RMSE between predicted and actual ratings. Apart from BiMax, no other biclustering algorithm is able to produce efficient recommendations. Furthermore, the coverage covered by BiMax algorithm seems to be very good. These results show that proposed algorithm has performed efficiently well on the given dataset.

## E. Result analysis on FilmTrust dataset

We performed experiments to measure the impact of used Biclustering algorithms on Film-Trust dataset. Figs 11 and 12 show values of various evaluation metrics while applying q-

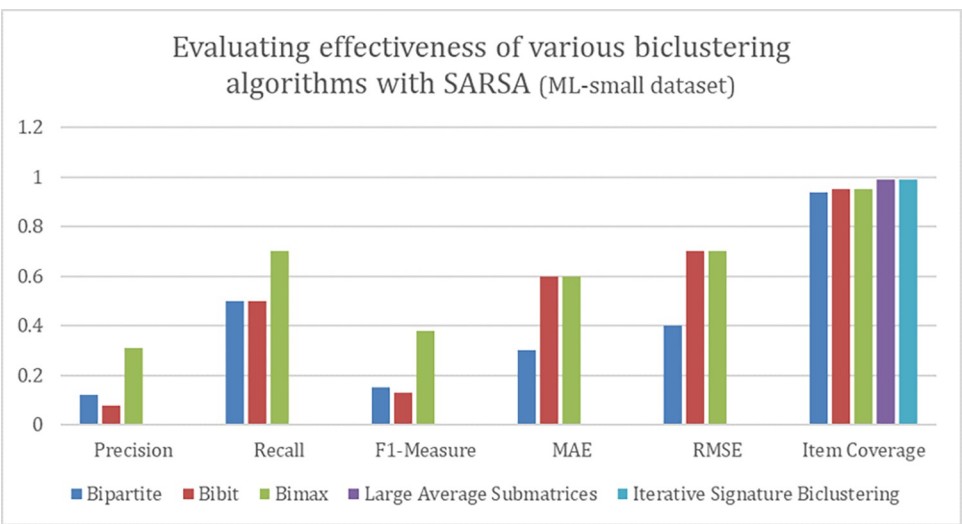

**Fig 10. Performance of various biclustering algorithms on ML-latest-small dataset with SARSA.**

**Table 7. Evaluation metrics with Q-learning on ML-latest-small.**

| Top n items | Personalization | Intra List Similarity | Novelty |
|---|---|---|---|
| **Bipartite** | 0.2 | 0.35 | 1.8 |
| **Bibit** | 0.5 | 0.21 | 3.7 |
| **BiMax** | 0.3 | 0.25 | 5 |
| **Least Average Submatrices** | - - - | - - - | - - - |
| **Iterative Signature Biclustering Algorithm** | - - - | - - - | - - - |

learning and SARSA to the given dataset. After analyzing results, it can be seen that BiMax biclustering algorithm has generated efficient recommendations. Moreover, the score of all evaluation metrics found to be good with regard to BiMax. Apart from BiMax, no other biclustering algorithm is able to produce efficient recommendations. Furthermore, the coverage covered by BiMax algorithm seems to be very good. One important point regarding FilmTrust dataset is that it lacks movies information. Without movie information, metrics like intra-list similarity, diversity, and personalization cannot be calculated as they rely on item attributes or features. For that very reason, we have not calculated values for above evaluation metrics for the FilmTrust dataset.

## F. Comparison with existing systems

To evaluate the effectiveness of proposed work, we performed comparison with state of the art existing techniques for generating recommendations for users. The unique aspect of proposed work is that proposed algorithms' effectiveness has been measured across various dimensions using multiple evaluation metrics. This enables us to inline effectiveness of RL based recommender system with new dimensions that are now being linked to recommender system in recent times like diversity, novelty and personalization for its effectiveness. Table 9 presents the results obtained after applying proposed algorithm on ML-100K dataset. The values obtained for precision, recall and F1-measure are better than most of the existing techniques. Moreover, on contrary to existing RL based recommendation system proposed work is able to predict to MAE and RMSE for the given dataset effectively using a novel approach. However, one important aspect that is to be considered is that proposed algorithm underlines low coverage in regard to generated recommendations. The high values of precision and recall highlights that the recommender system excels at finding and proposing products that fit consumers' tastes. A benefit of the system's performance is that users are getting recommendations that are accurate and pertinent. Item coverage also proved to be better than RLRS1 and RLRS2. Moreover, on contrary to existing systems, proposed algorithm has been evaluated for diversity, personalization and novelty. The values obtained for these metrics also highlights the effectiveness of proposed algorithm.

Table 10 presents results on FilmTrust dataset for competitor methods. MAE and RMSE prove to best for the proposed algorithm. Precision, Recall and F1 measures are better than all

**Table 8. Evaluation metrics with SARSA on ML-latest-small.**

| Algorithms | Personalization | Intra List Similarity | Novelty |
|---|---|---|---|
| **Bipartite** | 0.3 | 0.4 | 1.9 |
| **Bibit** | 0.4 | 0.3 | 8 |
| **BiMax** | 0.2 | 0.25 | 5 |
| **Least Average Submatrice Clustering Technique** | - - - | - - - | - - - |
| **Iterative Signature Biclustering Algorithm** | - - - | - - - | - - - |

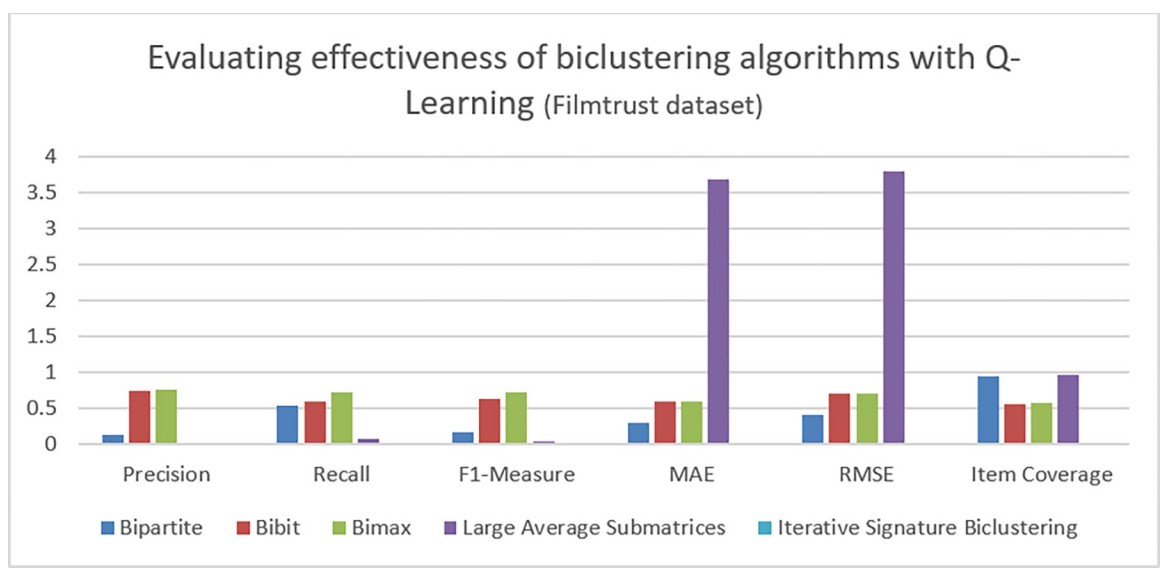

**Fig 11. Performance of various biclustering algorithms on FilmTrust dataset with Q-learning.**

existing techniques apart from RLRS1 and RLRS2. However, one major problem of existing RL based recommender systems is that they cannot predict MAE and RMSE. Our proposed algorithm has been able to overcome this deficiency and been able to predict MAE and RMSE with good accuracy. Moreover, the item coverage seems to be very good for this dataset. The main reason is that dataset is not sparse and relevant items are being effective retrieved from set of biclusters generated by biclustering algorithms. One major drawback of FilmTrust dataset is that it does not contain demographic information of movies. This information is essential to calculate values for evaluation metrics like intra list similarity and novelty. For that reason, values of these evaluation metrics cannot be obtained for the FilmTrust dataset. However, personalization value seems to be very good for the proposed algorithm.

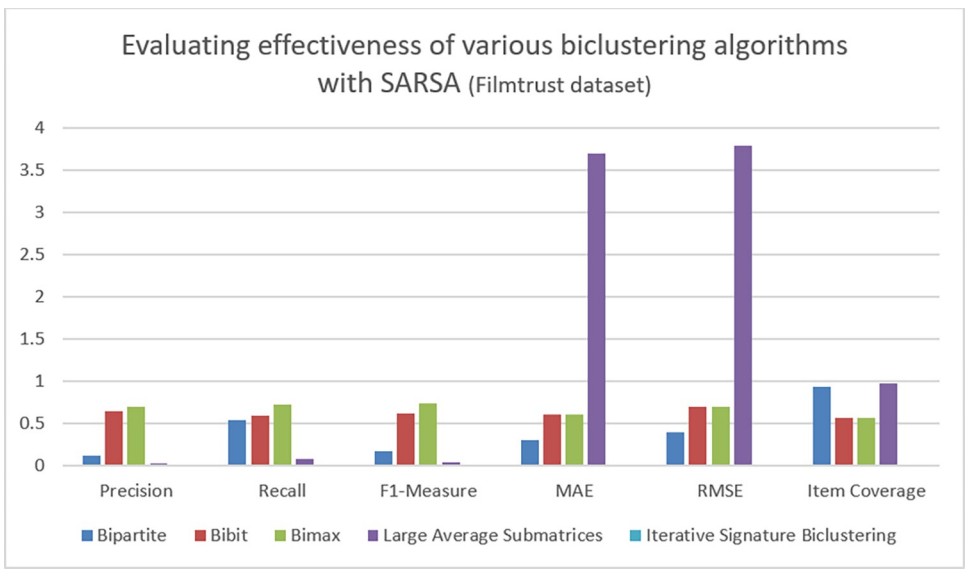

**Fig 12. Performance of various biclustering algorithms on FilmTrust dataset with SARSA.**

**Table 9. Competitor methods results on ML-100K dataset.**

|  | IPWR | ITR | SPOP | S4 | RLRS1 | RLRS2 | Proposed |
|---|---|---|---|---|---|---|---|
| **MAE** | 0.814 | 0.825 | 0.991 | 1.015 | - - - | - - - | 1.042 |
| **RMSE** | 1.029 | 1.039 | 1.245 | 1.315 | - - - | - - - | 1.207 |
| **Precision** | 0.615 | 0.616 | 0.477 | 0.491 | 0.425 | 0.52 | 0.50 |
| **Recall** | 0.349 | 0.336 | 0.412 | 0.405 | 0.691 | 0.75 | 0.80 |
| **F1-measure** | 0.411 | 0.402 | 0.404 | 0.410 | 0.518 | 0.61 | 0.61 |
| **Item Coverage** | 100 | 100 | 92.6 | 95.2 | 68.89 | 65.45 | 71.23 |
| **Personalization** | - - - - - - | - - - - - - | - - - - - - | - - - - - - | - - - - - - | 0.55 | 0.6 |
| **Intra list Similarity** | - - - - - - | - - - - - - | - - - - - - | - - - - - - | - - - - - - | 0.4 | 0.29 |
| **Novelty** | - - - - - - | - - - - - - | - - - - - - | - - - - - - | - - - - - - | 4.2 | 3.5 |

Table 11 presents results for ML-Latest -Small dataset. This dataset is not being extensively used in the field of recommender system. The main reason is due to its sparsity and lack of benchmarking. However, it can give us the insight regarding the efficiency of proposed algorithm. The values obtained for precision, recall and F-measure results are best as compared to limited recommendation techniques that have been applied to ML-latest-small dataset. The high value of item coverage for proposed method underlines that recommendation algorithm generated relatively good number of recommendations but due to sparsity of data only limited number of items are relevant to user which makes precision score low. Moreover, RMSE and personalization score seems good for the proposed system. Moreover, high value of novelty indicates that proposed algorithm is being able to recommend novel items to users. The overall score of evaluation metrics underline that proposed algorithm has effectively generated recommendations for the given sparse dataset as compared to existing techniques.

After applying proposed algorithm on three datasets, it can be seen that proposed algorithm has performed very well on all of datasets. This highlights the adaptability of proposed algorithm as it effectively captured varying users' interest across multiple datasets. Moreover, it further highlights the generalization and scalability of proposed algorithm as it has been applied to diverse datasets of varying context. The generated recommendations have been of good quality which have been evaluated across various spectrums in context of recommendation systems.

## VI. Conclusion and future work

Enormous increase in online data, is resulting in growing market of RSs. Thus, having a more accurate and context aware RS is need of the time. The availability of vast amount of digital

**Table 10. Competitor methods results on FilmTrust dataset.**

|  | IPWR | ITR | SPOP | S4 | RLRS1 | RLRS2 | Proposed |
|---|---|---|---|---|---|---|---|
| **MAE** | 0.637 | 0.625 | 0.655 | 0.635 | - - - | - - - | 0.601 |
| **RMSE** | 0.993 | 0.83 | 0.86 | 0.83 | - - - | - - - | 0.712 |
| **Precision** | 0.605 | 0.62 | 0.55 | 0.54 | 0.725 | 0.75 | 0.78 |
| **Recall** | 0.545 | 0.51 | 0.494 | 0.515 | 0.849 | 0.72 | 0.95 |
| **Fmeasure** | 0.578 | 0.57 | 0.519 | 0.526 | 0.765 | 0.73 | 0.80 |
| **Item Coverage** | 98.2 | 99.63 | 96.1 | 96.1 | 89.07 | 95.6 | 96.2 |
| **Personalization** | - - - - - - | - - - - - - | - - - - - - | - - - - - - | - - - - - - | - - - - - - | 0.5 |
| **Intra list Similarity** | - - - - - - | - - - - - - | - - - - - - | - - - - - - | - - - - - - | - - - - - - | - - - - - - |
| **Novelty** | - - - - - - | - - - - - - | - - - - - - | - - - - - - | - - - - - - | - - - - - - | - - - - - - - - |

**Table 11. Competitor methods results on ML-latest-small dataset.**

| | ITR | IPWR | SPOP | S4 | RLRS1 | RLRS2 | Proposed |
|---|---|---|---|---|---|---|---|
| MAE | 0.662 | 0.686 | 0.71 | 0.68 | ---- | ---- | 0.61 |
| RMSE | 0.872 | 0.910 | 0.76 | 0.71 | ---- | ---- | 0.72 |
| Precision | 0.617 | 0.608 | 0.32 | 0.35 | 0.25 | 0.31 | 0.42 |
| Recall | 0.402 | 0.411 | 0.49 | 0.51 | 0.07 | 0.09 | 0.71 |
| Fmeasure | 0.485 | 0.489 | 0.38 | 0.41 | 0.11 | 0.11 | 0.52 |
| Item Coverage | 98.12 | 95.23 | 96.1 | 96.1 | 0.12 | 90.0 | 95.0 |
| Personalization | ------- | ------- | ------- | ------- | ---- | 0.30 | 0.35 |
| Intra list Similarity | ------- | ------- | ------- | ------- | ---- | 0.21 | 0.25 |
| Novelty | ------- | ------- | ------- | ------- | ---- | 4.6 | 5.0 |

information on internet makes RSs as a potential market for effectively providing context aware and relevant information to the users. However, user's taste keeps on changing with time which implies that recommendation process should be dynamic in nature so that it can cope with changing user's preferences. In this work, recommendation problem has been presented as MDP problem, where RL agent traverses on a squared grid presented as an environment for the RL agent. During the traversal, RL agent generate recommendations for the active user. However, one major problem with RL is its high computational cost because RL agent has to traverse whole user-item ratings matrix to generate recommendations. To overcome this problem, Biclustering is applied to generate biclusters before feeding the data to RL agent. The merger of Biclustering and RL helps to create scalable and personalized recommendations. To understand utility of various Biclustering algorithms, we applied eight different biclustering algorithms namely, Plaid, Qubic, Cheng and Church, Bibit, BiMax, Bipartite, Large Average Submatrix and Iterative signature biclustering algorithm on used datasets. We observed that three algorithms Plaid, Qubic and Cheng and Church are not suitable for recommendation problem. As these algorithms generated biclusters that were comparable to the size of original dataset thus failing to the objective of extracting local hidden correlations. Remaining five biclustering algorithms Bibit, BiMax, Bipartite, Large Average Submatrix and Iterative signature biclustering algorithm have been successfully applied to the used recommendation datasets. This helped us in further understanding that which type of biclustering algorithm is best suited for which sort of recommendation task. Generated biclusters of these algorithms are then mapped to 6×6 square grid, which is then used as an environment for the RL agent. Due to finite size of grid, top $n^2$ biclusters are being selected for placement on the squared grid. Afterward, RL agent moves on squared grid and generate recommendations. Moreover, opposed to existing RL based recommendation methods that only predict items, we presented a novel approach to predict rating of the predicted items. Predicted rating helped us to calculate MAE and RMSE for the recommended items. Furthermore, recommendation quality is measured by using some new quality metrics. The values obtained for these evaluation metrics highlights that the proposed algorithm has produced personalized recommendation for all of the three datasets used. Additionally, via experimentation it is found that Bibit and BiMax are two biclustering algorithms that produced best results in generating recommendations for all used datasets. This finding can help other researchers of this field in a way that they should choose only BiMax and Bibit instead of trying other biclustering algorithms. The use of biclustering techniques enable proposed algorithm to give scalable and personalized recommendations with changing user preferences. In future, we intend to explore more of the unique ways to construct and traverse environment for RL based recommender systems. This may involve incorporating user's demographic information in computing reward function. Increase in

action space of the agent to explore environment. Furthermore, an enhance quality measure may be developed to measure the quality of biclusters. Moreover, some novel techniques may be identified to handle those items which may not become part of any bicluster and remain inactive while generating recommendations for the active user. The cold start problem is another problem that may be addressed for RL based recommendation via exploration. In the conclusion, this work's findings not only advance our knowledge of efficient biclustering algorithms in recommendation systems, but also open new avenues for future research to fully realize reinforcement learning's potential for producing excellent, context-aware recommendations.

## Author Contributions

**Conceptualization:** Muhammad Waqar, Mubbashir Ayub.

**Data curation:** Muhammad Waqar, Mubbashir Ayub.

**Formal analysis:** Muhammad Waqar.

**Investigation:** Muhammad Waqar.

**Methodology:** Muhammad Waqar.

**Supervision:** Mubbashir Ayub.

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
