## [Decision Letter · Decision Letter 0]

9 Oct 2024

PONE-D-24-18432A Personalized Reinforcement Learning Recommendation Algorithm using Bi-Clustering TechniquesPLOS ONE

Dear Dr. Ayub,

Thank you for submitting your manuscript to PLOS ONE. After careful consideration, we feel that it has merit but does not fully meet PLOS ONE’s publication criteria as it currently stands. Therefore, we invite you to submit a revised version of the manuscript that addresses the points raised during the review process.

We look forward to receiving your revised manuscript.

Kind regards,

Yunhe Wang

Academic Editor

PLOS ONE

Journal Requirements:

4. Thank you for stating the following in your Competing Interests section:  No Competing interest.

5. In the online submission form, you indicated that data can be provided on request

Reviewers' comments:

Reviewer's Responses to Questions

**Comments to the Author**

1. Is the manuscript technically sound, and do the data support the conclusions?

Reviewer #1: Yes

Reviewer #2: Yes

2. Has the statistical analysis been performed appropriately and rigorously? 

Reviewer #1: Yes

Reviewer #2: Yes

3. Have the authors made all data underlying the findings in their manuscript fully available?

Reviewer #1: Yes

Reviewer #2: Yes

4. Is the manuscript presented in an intelligible fashion and written in standard English?

Reviewer #1: Yes

Reviewer #2: Yes

5. Review Comments to the Author

Reviewer #1: Reviewer#01

The paper presents an excellent dive into the application of biclustering and Reinforcement Learning (RL) in the domain of Recommender Systems. But the domain of Recommender Systems is very vast and authors didn’t mention the application domain of proposed system. From Datasets it gives an illusion that proposed method can be applied to movies prediction domain. Authors implemented and tested eight biclustering algorithms and identified their potential applicability to the movie Recommender Systems domain, which can be helpful for future research. A rating prediction mechanism is also suggested by authors to predict movies rating, which also seems to be novel. Overall work is good and my recommendation is accept with following minor revisions to be incorporated.

Q#01: Abstract should be more concrete shedding more light on the novelties of your work.

Q#02: the datasets used for experimentation should be described in the abstract.

Q#03: In which domain this research can be used?

Q#04: Give motivation of this research also in abstract?

Q#05: At line 44,45 “Moreover, with increased size of data, these algorithms’ performance decreases gradually”. Any reference of this? Or it is your own statement?

Q#06: At line 46, 47 give reference of 1 or 2 relevant works and drawbacks. Rather than starting In this work we are…..

Q#07: Line 48, 49 “In this work we are combining the benefits of biclustering with power of RL to generate personalized

recommendations”. Should be placed in abstract for readers’ interest.

Q#08: Why there is a need to predict rating in RL based Recommender Systems?

Q#09: At line 58,59 Biclusters on a squared grid prove to be a cost effective solution while traversing through environment. More appropriately Change it to “can be a cost effective….”

Q#10: Figure 2 methodology should be more appropriate depicting your core work and more specific. Not generalized.

Q#11: Literature review should be relevant and precise not very broad. Like knowledge based systems, Context aware systems, Social network based systems be removed.

Q#12: A more conclusive statement needs to be added in conclusion and future work section.

Q#13: All figures quality needs to be improved.

Q#14: Line 417, 429, 438, 449, 459. The discussion part of algorithms should be in Literature review

Q#15: Line 1140 to Line 1146, make this paragraph more meaningful and specific. Don’t use generic terms.

Reviewer #2: In this work authors presented the application of biclustering in the domain of Recommender Systems. Very limited work is done in literature regarding the application of biclustering in the domain of Recommender Systems. In this regard work of authors is novel as authors verified the applicability of biclustering in the domain of recommender systems. Eight biclustering algorithms are applied to three datasets of movie domain. Generated biclusters are then used for state representation of the environment for an RL agent. RL agent is aimed to find a set of relevant items for the target user. Authors also presented a novel way to predict ratings of relevant items. Work is good but not presented in a good manner. Several typos and grammatical mistakes of English language are found in the manuscript. My recommendation is to correct these mistakes before final acceptance of the manuscript. Following should be corrected.

1. Abstract should be more concrete highlighting what you have done not what others have done.

2. Notations are not identical. For example, in Eq. (2), it is mentioned that, A, represents a bicluster but at several places like in figure 1, B, is used to denote a bicluster.

3. Several grammatical mistakes like Let assume should be Let’s assume.

4. Literature review should be relevant and precise not very broad. Like knowledge based systems, Context aware systems, Social network based systems be removed.

5. In section IV, you mentioned Table 4 shows representations of various grid positions of generated biclusters after applying various biclustering algorithms. Instead of word various write names of biclustering algorithm for more clarity.

6. Name of ML-Latest-Small dataset is not consistent in the whole manuscript.

7. Authors mentioned in section V that RLRS1 [53] used only one biclustering algorithm to generate Biclusters, But which one? Name it.

8. Also name algorithms that RLRS2 used to generate biclusters and which quality measure was used?

9. Figure 4(b), shows the steps taken by agent while exploring squared grid for ML-100K dataset, but authors mentioned it as figure 5(b).

10. Name of Large Average Submatrix Biclustering Algorithm is confused at many places, in section III(B)(4) it is Large Average Value Biclustering Algorithm. In table 7 it is Least Average Submatrices. What is this? Are they same or different?

11. Conclusion and future work section should be more elaborated.

6. PLOS authors have the option to publish the peer review history of their article (what does this mean?). If published, this will include your full peer review and any attached files.

Reviewer #1: **Yes: **Mustafa Bin Tariq

Reviewer #2: No

---

## [Author Response · Author response to Decision Letter 0]

11 Nov 2024

Dear Editor,

Thank you for allowing a resubmission of our manuscript, with an opportunity to address the reviewers’ comments.

We are uploading (a) our point-by-point response to the comments (below) (response to reviewers), (b) an updated manuscript with yellow highlighting indicating changes, and (c) a clean updated manuscript without highlights.

---

## [Decision Letter · Decision Letter 1]

27 Nov 2024

A Personalized Reinforcement Learning Recommendation Algorithm using Bi-Clustering Techniques

PONE-D-24-18432R1

Dear Dr. Ayub,

We’re pleased to inform you that your manuscript has been judged scientifically suitable for publication and will be formally accepted for publication once it meets all outstanding technical requirements.

Kind regards,

Yunhe Wang

Academic Editor

PLOS ONE

Additional Editor Comments (optional):

Reviewers' comments:

Reviewer's Responses to Questions

**Comments to the Author**

1. If the authors have adequately addressed your comments raised in a previous round of review and you feel that this manuscript is now acceptable for publication, you may indicate that here to bypass the “Comments to the Author” section, enter your conflict of interest statement in the “Confidential to Editor” section, and submit your "Accept" recommendation.

Reviewer #1: All comments have been addressed

Reviewer #2: All comments have been addressed

2. Is the manuscript technically sound, and do the data support the conclusions?

Reviewer #1: Yes

Reviewer #2: Yes

3. Has the statistical analysis been performed appropriately and rigorously? 

Reviewer #1: Yes

Reviewer #2: Yes

4. Have the authors made all data underlying the findings in their manuscript fully available?

Reviewer #1: Yes

Reviewer #2: Yes

5. Is the manuscript presented in an intelligible fashion and written in standard English?

Reviewer #1: Yes

Reviewer #2: Yes

6. Review Comments to the Author

Reviewer #1: (No Response)

Reviewer #2: Authors have answered all of the questions. I think this version of the manuscript can be accepted for publication.

7. PLOS authors have the option to publish the peer review history of their article (what does this mean?). If published, this will include your full peer review and any attached files.

Reviewer #1: **Yes: **Mustafa Bin Tariq

Reviewer #2: No

---

## [Editor Report · Acceptance letter]

16 Dec 2024

PONE-D-24-18432R1 

PLOS ONE

Dear Dr. Ayub, 

I'm pleased to inform you that your manuscript has been deemed suitable for publication in PLOS ONE. Congratulations! Your manuscript is now being handed over to our production team.

Kind regards, 

on behalf of

Dr. Yunhe Wang 

Academic Editor

PLOS ONE